# Compressing Large MoE Models via Efficient Pruning and Data-Aware Calibration

## Abstract

Ultra-large Mixture-of-Experts (MoE) language models, *e.g.,* DeepSeek-R1, are rapidly emerging as a dominant architecture due to their superior scalability and performance. However, the massive number of expert parameters introduces substantial redundancy, posing serious challenges for efficient deployment. Existing pruning methods face two fundamental challenges when applied to such MoE architectures. First, while methods based on reconstruction loss offer a more comprehensive selection by considering each expert combination, the vast search space renders exhaustive evaluation infeasible. Second, most approaches rely on a fixed calibration dataset to guide pruning, which often fails to preserve the model's full capabilities. To address these challenges, we introduce two key innovations in our pruning framework. First, we propose a *Coarse-to-Fine Expert Selection* strategy that reduces the computational complexity of reconstruction-loss–based selection from an exponential ($\mathcal{O}(\binom{2n}{n})$) to a polynomial scale ($\mathcal{O}(n^{1.5})$) with respect to the number of experts. This significantly accelerates the pruning process without sacrificing selection quality. Second, we develop a *Dynamic Calibration Dataset Mixing* strategy that enables the model to adaptively adjust its calibration set during pruning. Extensive experiments on a range of benchmarks show that our method can prune 50% of the experts in a large-scale MoE model (*e.g.,* DeepSeek-R1) while retaining 98.9% of its original performance across diverse tasks, outperforming existing pruning baselines. Our approach also demonstrates practical speedups and reduced memory footprint, facilitating efficient real-world deployment. The anonymous implementation is available at `https://anonymous.4open.science/r/DCDM-4C65-622a2bad88498795b8d7a92d85aca1315f9520ee`.

## 1 Introduction

Mixture-of-Experts (MoE) models (Zhao et al., 2023; Team, 2024b; Jiang et al., 2024; Dai et al., 2024a; Fedus et al., 2022) have demonstrated remarkable performance across a wide range of natural language processing tasks due to their ability to scale model capacity via a large number of experts while keeping per-token computation relatively constant by activating only a small subset at each step. However, a major bottleneck in deploying MoE models is the memory overhead introduced by inactive experts (Gao et al., 2022), which must still be stored during inference despite not contributing to computation. This issue is particularly severe for ultra-large models with a massive pool of experts (DeepSeek-AI et al., 2025; Team, 2025), of which only a small fraction are ever activated. If inactive experts could be effectively removed or reduced, the memory and storage costs would drop significantly, making MoE models much more practical for real-world deployment.

A straightforward and widely used approach is to prune experts based on router-derived metrics, such as gate values or activation frequency (Cao et al., 2024b; Dong et al., 2025). This strategy is computationally inexpensive and requires only a single forward pass, making it attractive for large-scale models. However, such metrics treat each expert independently and fail to account for the mutual influence among experts, often leading to suboptimal pruning decisions. To more accurately measure an expert's contribution, some work introduces reconstruction-loss-based selection (Lu et al., 2024), which evaluates the change in model outputs when an expert is removed. Although this provides a more principled criterion, it typically requires evaluating a large number of

expert combinations, which becomes computationally prohibitive as the number of experts grows. To further reduce redundancy while preserving knowledge, clustering-based methods (Chen et al., 2025) attempt to merge similar experts, yet these approaches still incur high computational cost and degrade performance in ultra-large models. Despite these advances, designing a pruning strategy that balances accurate expert contribution estimation with computational efficiency remains a key challenge that we aim to address in this work.

To address these challenges, we introduce a pruning framework for MoE models grounded in reconstruction-loss-based selection. First, to alleviate the massive computation induced by huge number of experts, we adopt a ***layer-wise greedy search*** that incrementally selects critical experts in each layer by minimizing the discrepancy between the outputs of the original and pruned layers. This design reduces the complexity from exponential, $\mathcal{O}(\binom{2n}{n})$, to polynomial, $\mathcal{O}(n^2)$, and can be further accelerated by a ***coarse-to-fine expert selection mechanism*** that lowers the cost to $\mathcal{O}(n^{1.5})$, enabling pruning even for ultra-large MoE models. We also provide a theoretical guarantee that the global error can be bounded by the accumulated layer-wise error, ensuring alignment with the overall objective. Second, to overcome the reliance on domain-specific calibration datasets, we introduce a ***dynamic calibration dataset mixing*** strategy that adaptively adjusts the mixture of samples from different domains based on the discrepancy between the original and pruned models, thereby enhancing the generalization of the pruned model across diverse domains. Together, these two innovations enable scalable and generalizable pruning for MoE models, making them far more practical for real-world deployment.

Extensive experiments validate the effectiveness of our framework. We evaluate the pruned models on math (*i.e.,* AIME) and code (*i.e.,* LiveCodeBench) benchmark, our pruned model achieves up to 98.9% of the original model's performance while reducing memory usage by 50%. Notably, it remains competitive even on challenging multi-domain scenarios, where existing pruning methods often suffer significant degradation. These results highlight that our approach not only scales to ultra-large MoE models but also preserves generalization across diverse domains, making it significantly more deployment-friendly in real-world settings.

## 2 RELATED WORK

In this section, we first trace the evolution of MoE architectures that enable large-scale language models to scale efficiently via sparse expert activation. Second, we examine the major bottleneck in MoE deployment, the memory overhead caused by inactive experts, and review recent efforts that aim to prune expert parameters to reduce deployment cost.

### 2.1 EFFICIENT MoE ARCHITECTURES

The introduction of sparsely activated Mixture-of-Experts (MoE) architectures has been a key driver in scaling large models. Early works such as GShard (Lepikhin et al., 2021) and the Switch Transformer (Fedus et al., 2022) demonstrated that activating only a small subset of experts per token can dramatically increase model capacity without proportionally increasing computation. Building on these foundations, the BASE Layers (Lewis et al., 2021) formulation modeled token–expert assignments as a constrained optimization problem, enabling balanced expert usage at scale. Meanwhile, Task-MoE (Kudugunta et al., 2021) extended the routing paradigm by assigning experts at the task level, reducing routing variance and improving efficiency in multilingual settings. More recent systems, including DeepSeek-MoE (Dai et al., 2024b) and Mixtral (Jiang et al., 2024), explored hierarchical routing and large-scale deployment settings, further improving throughput and stability. These architectural innovations collectively highlight the efficiency gains achieved through sparse expert activation, laying the groundwork for subsequent research on reducing redundancy among experts and optimizing their deployment. These designs typically require training from scratch. In contrast, a more practical challenge is how to reduce the number of experts without reinitializing or retraining the full model.

### 2.2 COMPRESSION OF MoE MODELS

Although sparse activation reduces computation, pretrained MoE models still incur substantial storage and memory overhead, motivating efforts to compress experts. Routing-based methods estimate

importance from token routing statistics (Lu et al., 2024; Chen et al., 2025) and are efficient but largely local, as they evaluate experts independently. Reconstruction-loss–based approaches (Cao et al., 2024a) provide a more principled functional criterion, yet existing formulations become prohibitively expensive for modern MoE layers with hundreds of experts, *e.g.,* directly computing reconstruction losses in DeepSeek-R1–scale models is computationally infeasible. We address this scalability gap by introducing a theoretically justified coarse-to-fine approximation that reduces reconstruction complexity from exponential to polynomial, making such pruning practical at large scale. A third line of work compresses MoE models by merging experts through clustering (Chen et al., 2025), though performance often degrades when scaling to highly diverse expert behaviors. Moreover, prior methods assume a fixed calibration dataset, overlooking the data-dependent nature of expert selection. While approaches such as DoReMi Xie et al. (2023a) dynamically reweight domain mixtures during training, pruning involves no gradient updates and must rely solely on forward-pass discrepancies. In contrast, we show that calibration composition significantly affects which experts should be retained and introduce a lightweight dynamic data-mixing mechanism that adapts this distribution during pruning without retraining.

## 3 PRELIMINARIES

**Notations and objective.** In general, we consider a Mixture-of-Experts (MoE) model with $L$ layers. Let $X_l$ and $X_{l+1}$ denote the input and output of the $l$-th layer, respectively. Each $\text{Layer}_l$ contains a router function $g_l$ that selects a subset of experts (*e.g.,* top-8 in DeepSeek-R1) from a total of $c$ experts. The full expert set in the $l$-th layer is denoted by $\mathcal{E}_{l,c}$, and any subset with $\hat{c}$ experts is written as $\mathcal{E}_{l,\hat{c}}$. The output of the $l$-th layer is given by $X_{l+1} = \text{Layer}_l(X_l; \mathcal{E}_{l,c})$. For notational brevity, we may simplify $\text{Layer}_l(\cdot; \mathcal{E}_{l,c})$ as $\text{Layer}_{l,c}$ when the input and expert set are clear from context. Then the overall model output is defined as:

$$X_{L+2} = \left(\text{Layer}_{L+1} \circ \text{Layer}_{L,c} \circ \cdots \circ \text{Layer}_{1,c}\right)(X_1), \tag{1}$$

where $\text{Layer}_{L+1}$ denotes the head layer. Let $\hat{X}_l$ and $\hat{X}_{l+1}$ be the input and output of the $l$-th pruned layer, then the output of the pruned model at the $l$-th layer is given by $\hat{X}_{l+1} = \text{Layer}_l(\hat{X}_l; \mathcal{E}_{l,\hat{c}}) := \text{Layer}_{l,\hat{c}}(\hat{X}_l)$, with $\hat{X}_1 = X_1$ as the input to the first MoE layer. The goal of pruning is to obtain, for each layer, a reduced expert subset $\mathcal{E}_{l,\hat{c}} \subset \mathcal{E}_{l,c}$ such that the output of the pruned model:

$$\hat{X}_{L+2} = \left(\text{Layer}_{L+1} \circ \text{Layer}_{L,\hat{c}} \circ \cdots \circ \text{Layer}_{1,\hat{c}}\right)(X_1), \tag{2}$$

approximates $X_{L+2}$ as closely as possible. Let $d(\cdot, \cdot)$ be a distance metric measuring the discrepancy between $X_{L+2}$ and $\hat{X}_{L+2}$. The pruning objective can then be formulated as:

$$\min_{\mathcal{E}_{l,\hat{c}}} d(X_{L+2}, \hat{X}_{L+2}), \quad \text{s.t.} \quad \mathcal{E}_{l,\hat{c}} \subset \mathcal{E}_{l,c}. \tag{3}$$

With this notation, we formalize expert pruning as selecting $\mathcal{E}_{l,\hat{c}}$ to minimize the output discrepancy. We next introduce our method for solving this problem efficiently.

## 4 METHODS

This section introduces our pruning framework, which primarily focuses on efficient expert selection and also incorporates a dynamic calibration strategy to improve cross-domain generalization. First, we present an expert selection method grounded in output discrepancy. We prove that the global pruning discrepancy can be bounded by the cumulative layer-wise discrepancy, providing a theoretical justification for adopting layer-wise pruning. Building on this analysis, we design a coarse-to-fine expert selection strategy (left side of Figure 1). Additionally, we introduce a dynamic calibration dataset mixing strategy (right side of Figure 1), which adaptively blends domain-specific and general data to further enhance the pruned model's generalization.

### 4.1 EXPERT SELECTION BASED ON DISCREPANCY

**Layer-wise greedy search.** The objective of pruning is to ensure that the output of the pruned model closely approximates that of the original model. However, directly optimizing expert selection across all layers to minimize the final output discrepancy is computationally impractical, as it

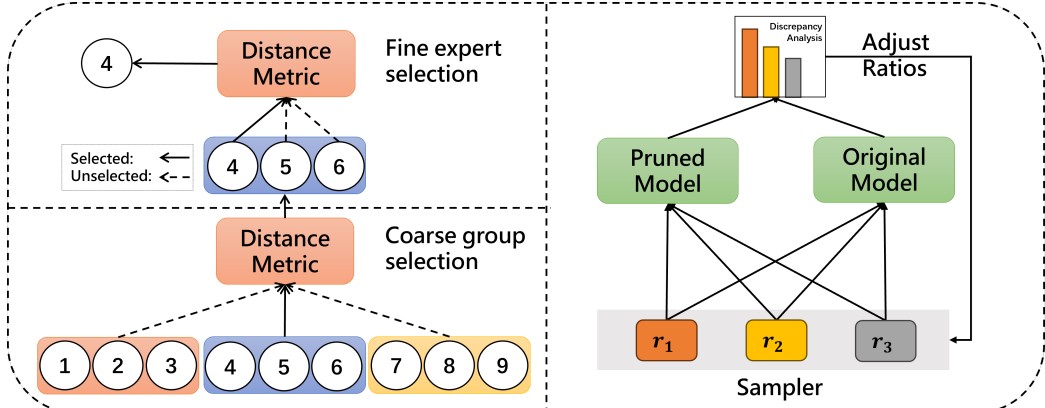

Figure 1: Overview of the proposed pruning framework. **Left**: Coarse-to-fine expert selection first scores groups of experts by output discrepancy, then refines the choice within the selected group. **Right**: Dynamic calibration dataset mixing adjusts domain sampling according to pruning-induced discrepancy, enhancing generalization across domains.

would require full forward passes through the model at every search step. A simpler alternative is to instead minimize the discrepancy between the original and pruned outputs at each layer individually. This raises a natural question:

> *Can the overall model output discrepancy be effectively controlled by minimizing layer-wise discrepancy?*

To address this, we propose the following theorem, which shows that the global output difference can indeed be bounded by the accumulation of local layer-wise discrepancy.

**Theorem 4.1** (Layer-wise Pruning Bound). *Let $X_{L+2}$ denote the output of the original model, $\hat{X}_{L+2}$ denote the output of the pruned model and $d(\cdot, \cdot)$ be a distance metric. Suppose each layer is pruned from $c$ experts to $\hat{c}$ experts. Then the distance $d(X_{L+2}, \hat{X}_{L+2})$ between the original and pruned outputs is bounded as follows:*

$$d(X_{L+2}, \hat{X}_{L+2}) \leq \sum_{i=2}^{L+1} \mathrm{Lip}_{i \to L+1} d(\hat{X}_i, \mathrm{Layer}_{i-1,c}(\hat{X}_{i-1}))  \tag{4}$$

*where $\mathrm{Lip}_{i \to L+1}$ denotes the Lipschitz constant from the $i$-th layer to the $(L+1)$-th layer.*

The complete proof is provided in Appendix B.

Theorem 4.1 shows that the discrepancy between the outputs of the original and the pruned model can be bounded by the accumulation of layer-wise differences. This result provides a theoretical justification for replacing the computationally expensive full-model comparison $d(X_{L+2}, \hat{X}_{L+2})$ with layer-wise comparisons $d(\hat{X}_i, \mathrm{Layer}_{i-1,c}(\hat{X}_{i-1}))$. Then, the costly process of repeatedly performing full forward passes and evaluating combinations of experts across multiple layers can be avoided, and the search space is reduced to experts within a single layer.

However, exhaustively evaluating all expert combinations within one layer still remains computationally expensive, exhibiting exponential time complexity (Lu et al., 2024) for layers with many experts (*e.g.,* 256 experts in DeepSeek-R1). Inspired by prior work (Cao et al., 2025), we adopt a greedy strategy that selects the most critical expert from each layer—the one that minimizes the output difference between the pruned and original models at a search time. This greedy approach reduces the time complexity from exponential to polynomial. Although polynomial complexity is more manageable, the computational cost can still be high during the search process with a large number of experts. To address this, we next introduce a *coarse-to-fine expert selection* strategy to further reduce the time cost.

---

**Algorithm 1** Coarse-to-Fine Expert Selection

---

**Require:**
1: **Calibration dataset**: $X_1$, **Selected experts**: $\hat{c}$.
2: **Architecture**: 1)Total layers $L$. 2) Experts per layer $N_e$.
3: **Experts initialization**:
- Selected experts at layer $l$: $\mathcal{E}_{l,\text{selected}} = \emptyset \quad (\forall l \in \{1, \ldots, L\})$.
- Candidate experts at layer $l$: $\mathcal{E}_{l,\text{candidate}} = \{1, 2, \ldots, N_e\} \quad (\forall l \in \{1, \ldots, L\})$.
- Group size: $S = \text{round}(\sqrt{N_e + \frac{1-\hat{c}}{2}})$.

**Ensure:** Pruned expert set $\{\mathcal{E}_{l,\text{selected}}\}_{l \in \{1,\ldots,L\}}$.
4: Compute each layer output of $X_{l+1}$ for each $\text{Layer}_l$ via calibration dataset $X_1$
5: **for** each MoE layer $l$ **to** $L$ **do**
6: $\quad \mathcal{E}_{\text{selected}} \leftarrow \mathcal{E}_{l,\text{selected}}, \mathcal{E}_{\text{candidate}} \leftarrow \mathcal{E}_{l,\text{candidate}}$
7: $\quad K \leftarrow \text{ceil}(|\mathcal{E}_{\text{candidate}}|/S)$
8: $\quad$ **for** $t = 1$ **to** $\hat{c}$ **do**
9: $\quad\quad$ Partition $\mathcal{E}_{\text{candidate}}$ into $K$ groups $\{\mathcal{G}_1, ..., \mathcal{G}_K\}$
10: $\quad\quad$ **for** each group $\mathcal{G}_k$ **do**
11: $\quad\quad\quad \mathcal{E}_{\text{temp}} \leftarrow \mathcal{E}_{\text{selected}} \cup \mathcal{G}_k$
12: $\quad\quad\quad$ Compute output distance $d_k \leftarrow d(X_{l+1}, \text{Layer}_l(X_l; \mathcal{E}_{\text{temp}}))$
13: $\quad\quad$ **end for**
14: $\quad\quad$ Find optimal group $\mathcal{G}^* \leftarrow \arg\min_{\mathcal{G}_k} d_k$
15: $\quad\quad$ Extract best expert $e^* \leftarrow \arg\min_{e \in \mathcal{G}^*} d(\text{Layer}_l(X_l; \mathcal{E}_{\text{selected}} \cup \{e\}), X_{l+1})$
16: $\quad\quad$ Update: $\mathcal{E}_{\text{selected}} \leftarrow \mathcal{E}_{\text{selected}} \cup \{e^*\}$
17: $\quad\quad$ Update: $\mathcal{E}_{\text{candidate}} \leftarrow \mathcal{E}_{\text{candidate}} \setminus \{e^*\}$
18: $\quad$ **end for**
19: $\quad \mathcal{E}_{l,\text{selected}} \leftarrow \mathcal{E}_{\text{selected}}$
20: **end for**

---

Table 1: Analysis of the time complexity.

| Methods | Time Complexity | Time Complexity in Terms of $n$ |
|---|---|---|
| O-Prune (Lu et al., 2024) | $\mathcal{O}\left(FL\binom{N_e}{\hat{c}}\right)$ | $\mathcal{O}(\binom{2n}{n})$ |
| HC-SMoE (Chen et al., 2025) | $\mathcal{O}(L(N_e - \hat{c})N_e^2 + FL)$ | $\mathcal{O}(n^3)$ |
| Layer-wise greedy search | $\mathcal{O}(\hat{c}FL(N_e + \frac{1-\hat{c}}{2}))$ | $\mathcal{O}(n^2)$ |
| Coarse-to-fine expert selection | $\mathcal{O}(\hat{c}FL\sqrt{4N_e + 2 - 2\hat{c}})$ | $\mathcal{O}(n^{1.5})$ |

**Coarse-to-fine expert selection**. To accelerate the otherwise expensive greedy search over all candidate experts, we adopt a coarse-to-fine strategy (illustrated on the left of Figure 1). The key idea is to first evaluate experts at a group level to quickly narrow down promising candidates, and then refine the choice within the selected group using a finer-grained metric. Concretely, the candidate expert set $\mathcal{E}_{l,c}$ at layer $l$ is first partitioned into $K$ disjoint groups $\mathcal{G}_1, \ldots, \mathcal{G}_K$, where the first $K - 1$ groups contain $S$ experts each and the last group holds the remaining candidates. The selection proceeds in two stages:

• **Coarse group selection**: For each group $\mathcal{G}_i$, we compute the output discrepancy between the original layer and a pruned layer that uses the currently selected experts $\mathcal{E}_{l,\text{selected}}$ together with all experts in $\mathcal{G}_i$. A distance metric (the $\ell_2$-norm in our experiments) measures this discrepancy. The group $\mathcal{G}^*$ with the smallest discrepancy is chosen as the most promising region.

• **Fine expert selection**: Within $\mathcal{G}^*$, we further evaluate each expert individually using the same distance metric and select the expert $e^*$ that yields the closest match to the original layer output when added to $\mathcal{E}_{l,\text{selected}}$.

The complete expert pruning algorithm is formally presented in Algorithm 1.

**Analysis of the time complexity.** We analyze the computational cost of our coarse-to-fine expert selection strategy in this part. Generally, we denote each MoE layer route to $N_e$ candidate experts and require $F$ time for a single forward pass. Then a naive greedy selection evaluates every remaining expert at each iteration, resulting in a large search space. Instead, our hierarchical strategy significantly shrinks this space. At the $t$-th iteration, the candidate pool is first reduced from $|N_e - t + 1|$

---

**Algorithm 2** Dynamic Calibration Dataset Mixing Strategy

---

**Require:** Initial weights $\mathbf{w}^0$, the required in Algorithm 1
1: **for** pruning iteration $t = 1$ to $T$ **do**
2:     Sample $\mathcal{X}_1$ from $\{D_1, \cdots, D_{N_D}\}$ with proportion $\boldsymbol{w}^{t-1}$.
3:     Prune model through Algorithm 1
4:     Evaluate the discrepancy $\boldsymbol{d}^t$ across $N_D$ domains
5:     Update weights: $\mathbf{w}^t \leftarrow \frac{\exp \boldsymbol{d}^t}{\|\exp \boldsymbol{d}^t\|_1}$
6:     **if** $\mathbf{w}^t == \mathbf{w}^{t-1}$ **then** break
7:     **end if**
8: **end for**

---

experts to $\lceil \frac{N_e - t + 1}{S} \rceil + S$. The resulting overall complexity is:

$$\mathcal{O}((\sum_{t=1}^{\hat{c}} \lceil \frac{N_e - t + 1}{S} \rceil + \hat{c}S)FL) = \mathcal{O}(\hat{c}FL(S + \frac{2N_e + 1 - \hat{c}}{2S})), \tag{5}$$

where the term $S + \frac{2N_e+1-\hat{c}}{2S}$ captures the trade-off between coarse grouping and fine expert selection. Then the optimal time complexity $\mathcal{O}(\hat{c}FL\sqrt{4N_e + 2 - 2\hat{c}})$ is achieved when the group size $S = \sqrt{N_e + \frac{1-\hat{c}}{2}}$, which is a significant improvement over the complexity of a naive greedy selection. We compare the time complexity of different methods in Table 1. By setting $N_e = 2n$ and $\hat{c} = n$, while treating other parameters as constants, the time complexity can be asymptotically expressed in terms of $n$.

To quantitatively assess the computational efficiency of our method relative to existing approaches, we examine the DeepSeek-R1 model as an example. It contains $L = 58$ layers with $N_e = 256$ routed experts per layer. When selecting $\hat{c} = 128$ experts, the method of O-Prune (Lu et al., 2024) would require $58\binom{256}{128}$ evaluations, HC-SMoE (Chen et al., 2025) about $4.87 \times 10^8$, and a layer-wise greedy search about $1.43 \times 10^6$. In contrast, our method needs only $2.06 \times 10^5$ evaluations, showing a substantial reduction in time cost and markedly better efficiency than competing approaches.

Overall, our coarse-to-fine expert selection scheme reduces computational cost and provides a practical approach for pruning large language models with numerous experts within a reasonable time.

## 4.2 DYNAMIC CALIBRATION DATASET MIXING STRATEGY

While existing work shows that domain-specific performance can often be well preserved, expert pruning still suffers from notable cross-domain degradation (Dong et al., 2025). To alleviate the domain shift introduced by the calibration data, we propose a Dynamic Calibration Dataset Mixing (DCDM) strategy, motivated by recent work (Xie et al., 2023b; Xia et al., 2024).

Specifically, given $N_d$ distinct domains $\{\mathcal{D}_i\}_{i=1}^{N_D}$, we initialize the domain mixing weights $\mathbf{w}^0 = (w_1^0, \ldots, w_{N_D}^0)$ proportionally to their dataset sizes $w_i^0 = \frac{|\mathcal{D}_i|}{\sum_{j=1}^{N_D} |\mathcal{D}_j|}$. Let $\boldsymbol{d}^t = (d_1^t, \ldots, d_{N_D}^t)$ denote the discrepancy between the original model and the pruned model outputs across $N_D$ domains at the $t$-th pruning iteration.

Accordingly, our objective is to minimize the performance gap between the pruned and original models, ensuring the outputs of the pruned model remains as close as possible to that of the original model. Specifically, if the discrepancy of domain $\mathcal{D}_i$ is larger, then increase the sample ratio of the $\mathcal{D}_i$ domain, otherwise, it is decreased. Formally, at the $t$-th pruning iteration the ratio is obtained by $\mathbf{w}^t = \frac{\exp(\boldsymbol{d}^t)}{\|\exp(\boldsymbol{d}^t)\|_1}$. And the dynamic calibration dataset mixing strategy is presented in Algorithm 2.

## 5 EXPERIMENTS

### 5.1 EXPERIMENT SETUP

**Base models.** We conduct expert pruning on two popular large MoE models: `DeepSeek-R1` and `Qwen3-30B-A3B-Thinking` (Team, 2025).

Table 2: Performance comparison with expert pruning on Qwen3-30B-A3B-Thinking.

| Dataset | Method | MMLU | Math500 | AIME25 | LCB | Average |
|---------|--------|------|---------|--------|-----|---------|
| - | Original | 78.59 | 96.12 | 85.00 | 66.00 | 81.43 |
| C4 | Weights | 46.99 | 14.57 | 0.42 | 0.00 | 15.50 |
| | HC-SMoE | 39.42 | 72.98 | 22.08 | 3.18 | 34.41 |
| | Ours | **59.91** | 3.73 | 0.42 | 0.00 | 16.02 |
| OpenR1-Math | Weights | 33.36 | 13.65 | 2.50 | 4.90 | 13.60 |
| | HC-SMoE | 47.06 | 73.20 | 20.00 | 32.00 | 43.07 |
| | Ours | 44.10 | 94.60 | 73.75 | 0.60 | 53.26 |
| rStar-Coder | Weights | 39.31 | 20.30 | 4.16 | 0.00 | 15.94 |
| | HC-SMoE | 47.05 | 72.08 | 24.58 | 2.40 | 36.53 |
| | Ours | 44.54 | 93.08 | 71.67 | **63.00** | 68.07 |
| Mixed Datasets | DCDM | 52.56 | **95.60** | **80.00** | 52.40 | **70.14** |

• **DeepSeek-R1.** It is a large-scale MoE model with 671B total parameters. It contains 61 layers, including 3 dense transformer layers and 58 MoE layers. Each MoE layer has 256 routed experts and 1 shared expert. During inference, the router selects 8 routed experts per layer, which are combined with the shared expert for computation.

• **Qwen3-30B-A3B-Thinking.** It is a 30.5B parameter MoE model with 48 MoE layers. Each layer contains 128 routed experts, with 8 experts selected per token during inference.

In our experiments, we apply a 50% sparsity ratio to the routed experts, reducing the number of experts per MoE layer from 256 to 128 for `DeepSeek-R1`, and from 128 to 64 for `Qwen3-30B-A3B-Thinking`.

**Baseline methods.** To evaluate the effectiveness of our method, we compare against two representative baselines chosen along two key dimensions: the pruning criterion (static statistics *vs.* output-aware perturbation) and the pruning strategy (expert pruning *vs.* expert merging). To enable a fair comparison, we carefully select baseline methods by varying one factor at a time, either the pruning criterion or the pruning mechanism, while keeping other variables such as the underlying model architecture and sparsity ratio consistent.

• **Weights (Routing-based):** Following CD-MoE (Cao et al., 2024b), experts are ranked by the ratio of average routing weight to activation frequency. This favors experts that are both frequently used and strongly weighted, mitigating bias toward rarely-used or weakly-activated ones. The top-$\hat{c}$ experts are retained.

• **HC-SMoE (Clustering-based):** HC-SMoE (Chen et al., 2025) merges semantically similar experts based on their impact on model outputs, rather than pruning them. This clustering-based strategy preserves output quality while reducing the expert count.

**Datasets.** Calibration data during pruning are sourced from three distinct task domains: C4 (Raffel et al., 2019) (knowledge), `OpenR1-Math` (LI et al., 2024; open r1, 2025) (mathematical reasoning), and `rStar-Coder` (Liu et al., 2025) (code synthesis). As for the calibration size, we use 32 calibration examples of approximately $4K$ tokens each under single-domain settings. In our dynamic calibration dataset mixing strategy, 32 examples are selected in total, with the domain mixture automatically adjusted based on the strategy's feedback. The pruned model's performance is evaluated on four established benchmarks: MMLU (Hendrycks et al., 2020) (knowledge), Math500 (Lightman et al., 2023) and AIME25 (math ai, 2025) (mathematical reasoning), and LiveCodeBench (Jain et al., 2024) (abbreviated "LCB"; code synthesis).

## 5.2 MAIN RESULTS

**Pruning results on Qwen3-30B-A3B-Thinking.** We evaluate our approach on the `Qwen3-30B-A3B-Thinking` model to analyze the effect of different calibration strategies, as summarized in Table 2. We first examine pruning with a calibration dataset drawn from a single domain. This setting often achieves strong in-domain performance but suffers from severe cross-domain degradation. For example, pruning with C4 data yields competitive results on MMLU (*i.e.,* 59.67) but fails on mathematical and coding tasks (*e.g.,* 0.42 on AIME25). Similarly, pruning

Table 3: Comparison of Data Mixing Strategies.

| Method | MMLU | Math500 | AIME25 | LCB | Average |
|--------|------|---------|--------|-----|---------|
| Fixed | 50.58 | 93.30 | 70.00 | 19.50 | 58.35 |
| DCDM | **52.56** | **95.60** | **80.00** | **52.40** | **70.14** |

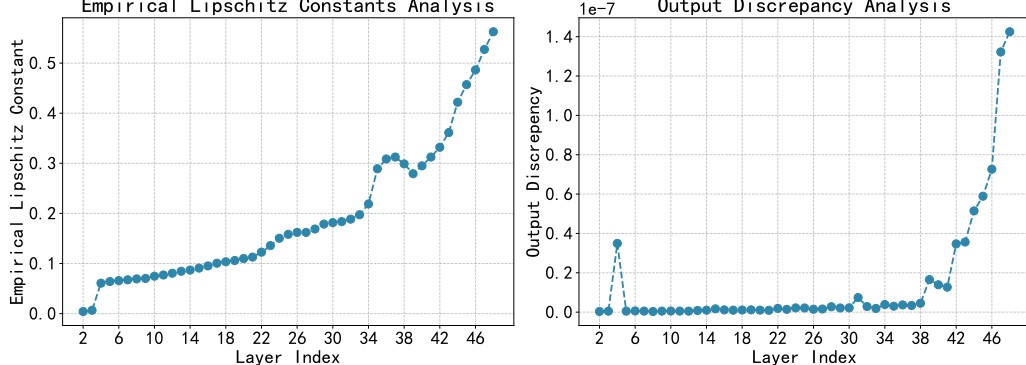

Figure 2: Empirical Lipschitz constants and output discrepancy versus layer index. (a) Lipschitz constants from the $i$-th layer to the 48th layer. (b) L2-norm between outputs of pruned and original models at each layer.

with domain-specific data such as OpenR1-Math excels in mathematics (*e.g.,* 94.60 on Math500) but generalizes poorly to other tasks (*e.g.,* 0.60 on LCB). These observations highlight a key limitation: single-domain pruning tends to overfit the calibration domain, causing catastrophic performance drops in other areas. To overcome this issue, we introduce the proposed Dynamic Calibration Dataset Mixing (DCDM) strategy, which adaptively reweights calibration data across domains according to pruning-induced output discrepancy. As shown in the final row of Table 2, DCDM achieves the highest average score (70.47), retaining 86.14% of the original model's performance (81.43) while outperforming all baselines. Notably, it avoids the severe performance collapses observed in single-domain pruning and delivers substantially better robustness and cross-domain generalization. Regarding whether mixed-domain calibration can yield higher peak performance than single-domain pruning, we provide a more detailed domain-level analysis in Appendix C.

### 5.3 EXPERIMENTAL ANALYSIS

**Empirical Analysis of Layer-wise Pruning Bound.** To empirically assess the tightness of the upper bound derived in Theorem 4.1, we estimate the empirical Lipschitz constants $Lip_{i \to L+1}$ for the Qwen3-30B-A3B-Thining model, which comprises 48 MoE layers. As shown in the left of Figure 2, most of estimated constants are below 0.5. This indicates that the coefficients $Lip_{i \to L+1}$ multiplying the layer-wise discrepancy in our theoretical bound are small across layers. The right side shows shows correspondingly small output discrepancies (most $\leq 1.4 \times 10^{-7}$) at each layer. These results provide empirical evidence for the tightness of our theoretical bound and support the effectiveness of the layer-wise pruning strategy.

**Time Cost vs. Number of Experts.** As shown on the left side of Figure 3, we measure the time cost per layer as the number of experts increases (128, 256, 512, and 1024). The plotted points represent actual measured costs, while the dashed line shows a fitted curve. The trend clearly demonstrates that the time cost of our method grows at a slower rate compared to other methods as the number of experts increases. This scaling behavior aligns with our time complexity analysis presented in Table 1, confirming that our approach remains efficient and practical for modern MoE models with large numbers of experts.

**Time Cost vs. Group Size.** The group size $S$ is a key hyper-parameter that balances the costs of the coarse and fine selection stages. We analyze its impact on the time cost for a fixed model size ($N_e = 128$ experts pruned to $\hat{c} = 64$), with results shown on the right side of Figure 3. Theoretically, the time complexity first decreases and then increases with $S$. Our empirical re-

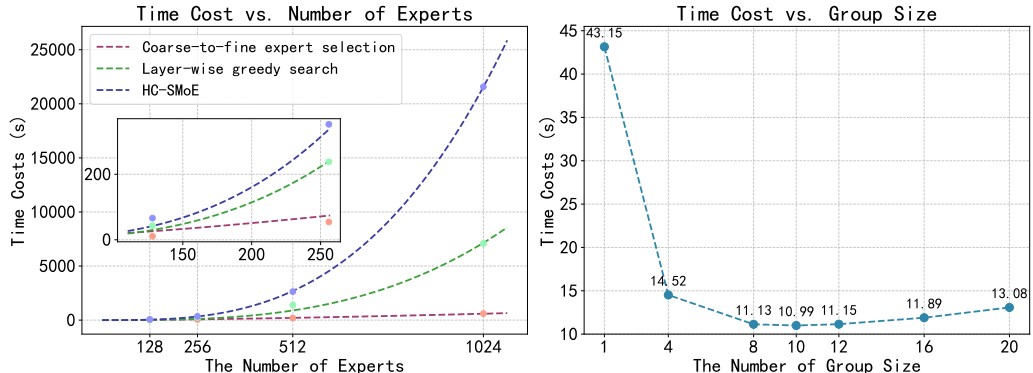

Figure 3: Time costs vs. number of experts and group size. Left: time cost versus the number of experts, with the inset providing a detailed perspective from 64 to 256 experts. Right: time cost versus group size.

sults strongly support this theoretical analysis. The measured time cost exhibits a characteristic U-shaped curve. The cost initially decreases sharply as $S$ increases from small values. It reaches a minimum around $S = 10$, which aligns closely with the theoretically predicted optimal value of $\text{round}(\sqrt{128 + (1 - 64)/2}) = 10$. Beyond this point, the cost increases gradually with further increases in $S$. The close agreement between theory and experiment demonstrates that our time complexity analysis is correct and identifies an optimal group size for minimizing pruning time.

**Data Mixing Strategies.** To evaluate our dynamic calibration dataset mixing (DCDM) strategy, we compare it with a static policy that combines rStar-Coder, OpenR1-Math, and C4 data in a fixed 1:1:1 ratio for the calibration set. As shown in Table 3, the fixed strategy performs reasonably well on math-related tasks (*e.g.,* 93.30 on Math500) but struggles on code-related tasks (19.50 on LCB). This indicates that a fixed ratio may not reflect the actual data needs of each domain, for instance, code tasks may require a higher proportion of domain-specific calibration samples. However, such requirements are typically unknown prior to pruning. Our DCDM strategy adjusts the calibration set composition based on pruning feedback, resulting in a modified ratio of 2:1:1. This adaptation increases the proportion of code domain data, which was underperforming with the 1:1:1 ratio. Consequently, our method improves performance across all domains, particularly on code tasks (19.50 vs. 58.10 on LCB). These results highlight the importance of calibration data composition in expert pruning and demonstrate that our dynamic approach can effectively balance data needs across domains to preserve overall model performance.

**Impact on Model Size.** To further evaluate the effectiveness and scalability of our approach on larger MoE models, we conduct experiments on the representative ultra-large model DeepSeek-R1, tested on two challenging benchmarks: AIME25 for math and LCB for code. The results, summarized in Table 4, show that the pruned models retain around 98.9% of the original model's performance while significantly reducing the number of active experts. Interestingly, we observe that larger models tend to maintain a higher fraction of their

Table 4: Performance on DeepSeek-R1.

| Method | AIME25 | LCB | Average |
|---|---|---|---|
| Original | 65.00 | 59.14 | 73.81 |
| Ours | 62.50 | 60.22 | 72.64 |

original performance after pruning, suggesting that scale improves robustness to compression. This observation is consistent with prior findings (Liu et al., 2024) that larger language models are generally more resilient to parameter reduction. These results highlight the scalability of our method and suggest promising potential for applying it to even larger future MoE models.

## 6 CONCLUSION

In this work, we propose an efficient expert pruning framework for large-scale Mixture-of-Experts (MoE) models, focusing on both pruning efficiency and performance preservation. Specifically, we introduce a coarse-to-fine polynomial selection strategy that reduces the search complexity from exponential scale to polynomial scale and a dynamic calibration data mixing strategy that adaptively adjusts calibration samples to improve the generalization among different domains. Experi-

ments across diverse domains and model scales show that our method surpasses existing baselines, achieving high compression rates while retaining up to 98.9% of the original model's performance, demonstrating its practicality and scalability for real-world MoE deployment.

## ETHICS STATEMENT

This work focuses on reducing memory cost of large-scale Mixture-of-Experts (MoE) models through expert pruning. Our research does not involve human subjects, personal data, or sensitive information, and all datasets used are publicly available and widely adopted in the community. We employed large language models only for grammatical checking and minor language polishing of the manuscript; they were not involved in designing experiments or generating results. We believe there is no foreseeable ethical risk beyond the general considerations of deploying more efficient large language models.

## REPRODUCIBILITY STATEMENT

We provide an anonymous implementation of our method to facilitate reproduction of all reported results. A detailed explanation of the proposed method and the proofs of theoretical justification is included in the Appendix B. The datasets and experimental settings are thoroughly described in Section 5.1, including datasets, baselines, and evaluation benchmarks, to ensure that readers can replicate our experiments without ambiguity. All hyperparameters and training configurations can be found in Section 5.1. At last, we used the public available pretrained models (*e.g.,* DeepSeek-R1 and Qwen3-30B-A3B-Thinking) for easy reproducibility.

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

# A  LLM USAGE

We use LLMs solely to check and correct grammatical errors in our paper.

# B  PROOFS

The goal of pruning is to minimize the difference between the output of the pruned model and that of the original model—that is, to minimize the distance $d(X_{L+2}, \hat{X}_{L+2})$. The proof of Theorem 4.1 proceeds in the following three steps.

*Proof.* We first derive a bound between the original and pruned models when pruning occurs in a single layer. For the $i$-th layer, where $\hat{c}$ experts are selected from the original $c$ experts, the output discrepancy can be bounded using the Lipschitz constants of the MoE layers from $i+1$ to $L$.

$$
\begin{aligned}
&d((\text{Layer}_{L,c} \circ \cdots \circ \text{Layer}_{i,c} \cdots \circ \text{Layer}_{1,c})(X_1), (\text{Layer}_{L,c} \circ \cdots \circ \text{Layer}_{i,\hat{c}} \cdots \circ \text{Layer}_{1,c})(X_1)) \\
&\leq \text{Lip}(\text{Layer}_{L,c} \circ \cdots \circ \text{Layer}_{i+1,c}) d(X_{i+1}, \text{Layer}_{i,\hat{c}}(X_i)).
\end{aligned}
\tag{6}
$$

In the second step, we drive the upper bound of the outputs distance when the number of layers is 3.

$$
\begin{aligned}
&d((\text{Layer}_{3,c}(X_2), (\text{Layer}_{3,c}(\hat{X}_2)) \\
&= d((\text{Layer}_{3,c} \circ \text{Layer}_{2,c} \circ \text{Layer}_{1,c})(X_1), (\text{Layer}_{3,c} \circ \text{Layer}_{2,\hat{c}} \circ \text{Layer}_{1,\hat{c}})(X_1)) \\
&\leq d((\text{Layer}_{3,c} \circ \text{Layer}_{2,c} \circ \text{Layer}_{1,c})(X_1), (\text{Layer}_{3,c} \circ \text{Layer}_{2,c} \circ \text{Layer}_{1,\hat{c}})(X_1)) \\
&\quad + d((\text{Layer}_{3,c} \circ \text{Layer}_{2,c} \circ \text{Layer}_{1,\hat{c}})(X_1), (\text{Layer}_{3,c} \circ \text{Layer}_{2,\hat{c}} \circ \text{Layer}_{1,\hat{c}})(X_1)) \\
&\leq \text{Lip}(\text{Layer}_{3,c} \circ \text{Layer}_{2,c}) d(\text{Layer}_{1,\hat{c}}(X_1), \text{Layer}_{1,c}(X_1)) \\
&\quad + \text{Lip}(\text{Layer}_{3,c}) d(\hat{X}_3, \text{Layer}_{2,c}(\hat{X}_2)) \\
&= \text{Lip}(\text{Layer}_{3,c} \circ \text{Layer}_{2,c}) d(\hat{X}_2, \text{Layer}_{1,c}(\hat{X}_1)) \\
&\quad + \text{Lip}(\text{Layer}_{3,c}) d(\hat{X}_3, \text{Layer}_{2,c}(\hat{X}_2)) \\
&= \sum_{i=2}^{3} \text{Lip}_{i \to 3} d(\hat{X}_i, \text{Layer}_{i-1,c}(\hat{X}_{i-1})),
\end{aligned}
\tag{7}
$$

where the first equality follows from the definitions of $X_2$ and $\hat{X}_2$. The second inequality holds according to the triangle inequality. And the third holds according to the inequality 6. The fourth equality follows from the definitions of $\hat{X}_2 = \text{Layer}_{1,\hat{c}}(X_1)$ and $\hat{X}_1 = X_1$. For convenience, we define $\text{Lip}_{i \to L}$ as the Lipschitz constant from the $i$-th layer to the $L$-th layer, which justifies the final equality.

In the third step, we generalize the result from 3 layers to the case of $K+1$ layers. Suppose the number of layers $K$, then there exists the following inequality holds

$$
d(\text{Layer}_{K,c}(X_K), \text{Layer}_{K,c}(\hat{X}_K)) \leq \sum_{i=2}^{K} \text{Lip}_{i \to K} d(\hat{X}_i, \text{Layer}_{i-1,c}(\hat{X}_{i-1}))
\tag{8}
$$

When the the number of layers is $K+1$, we have

$$
\begin{aligned}
&d(\text{Layer}_{K+1,c}(X_{K+1}), \text{Layer}_{K+1,c}(\hat{X}_{K+1})) \\
&\leq d(\text{Layer}_{K+1,c} \circ \text{Layer}_{K,c}(X_K), \text{Layer}_{K+1,c} \circ \text{Layer}_{K,c}(\hat{X}_K)) \\
&\quad + d(\text{Layer}_{K+1,c} \circ \text{Layer}_{K,c}(\hat{X}_K), \text{Layer}_{K+1,c} \circ \text{Layer}_{K,\hat{c}}(\hat{X}_K)) \\
&\leq \text{Lip}_{K+1} d(\text{Layer}_{K,c}(X_K), \text{Layer}_{K,c}(\hat{X}_K)) + \text{Lip}_{K+1} d(\hat{X}_{K+1}, \text{Layer}_{K,c}(\hat{X}_K)) \\
&= \sum_{i=2}^{K+1} \text{Lip}_{i \to K+1} d(\hat{X}_i, \text{Layer}_{i-1,c}(\hat{X}_i)),
\end{aligned}
\tag{9}
$$

where the first inequality holds according to the triangle inequality. The second inequality holds according to the Lipschitz constant of $(K + 1)$-th layer. The last inequality follows from inequality 8. Therefore, when the $K = L$, we have

$$d(X_{L+2}, \hat{X}_{L+2}) = \sum_{i=2}^{L+1} \text{Lip}_{i \to L+1} d(\hat{X}_i, \text{Layer}_{i-1,c}(\hat{X}_{i-1})) \tag{10}$$

$\square$

## C  EXPERIMENTAL ANALYSIS

**Grouping Strategies Analysis.** The grouping strategy in Algorithm 1 is based on expert order. We also evaluated two other strategies after pruning 50% of experts in Qwen3-30B-A3B-Thinking.

- **Random grouping:** experts are shuffle randomly before group.
- **Similarity-based grouping:** experts are grouped by greedily selecting the most similar candidates for each group.
- **Order-based grouping:** experts are grouped by their original indices.

As shown in Table 5, all grouping strategies preserve competitive performance, with order-based grouping slightly outperforming the others.

Table 5: Performance comparison of different grouping strategies.

| Method | MMLU | Math500 | AIME25 | LCB | Average |
|---|---|---|---|---|---|
| Original | 78.59 | 96.12 | 85.00 | 66.00 | 81.43 |
| Random | **52.59** | 92.20 | 73.30 | 47.10 | 66.30 |
| Similarity-based | 52.56 | 95.20 | **80.00** | 51.50 | 69.82 |
| Order-based | 52.56 | **95.60** | **80.00** | **52.40** | **70.14** |

**Distribution of Pruned Experts Analysis.** To illustrate the distribution of expert selection after pruning, we divide the remaining 64 experts into 8 groups of 16 experts each. As shown in Figure 4 5 6, the distribution of selected experts is uniform across different layers. We also compute the entropy of the router's output distribution over these groups. The results in Table 6 show that all entropy values are above 2.9 (with a maximum of 3.0), indicating a uniform expert selection without preference for any specific subset.

Table 6: Entropy of expert selection after 50% pruning.

| Layer Index | 6 | 12 | 18 | 24 | 30 | 36 | 42 | 48 |
|---|---|---|---|---|---|---|---|---|
| Order-based grouping | 2.91 | 2.97 | 2.93 | 2.95 | 2.97 | 2.95 | 2.94 | 2.98 |
| Random grouping | 2.94 | 2.96 | 2.95 | 2.98 | 2.99 | 2.95 | 2.94 | 2.98 |
| Similary-based grouping | 2.92 | 2.98 | 2.93 | 2.97 | 2.98 | 2.94 | 2.94 | 2.98 |

Table 7: Gini coefficients (original vs. pruned).

| Model | 6 | 12 | 18 | 24 | 30 | 36 | 42 | 48 |
|---|---|---|---|---|---|---|---|---|
| Original | 0.52 | 0.58 | 0.61 | 0.57 | 0.60 | 0.57 | 0.60 | 0.68 |
| Pruned | 0.43 | 0.49 | 0.54 | 0.53 | 0.55 | 0.50 | 0.50 | 0.54 |

**Analysis of Pruning Dataset Sequences.** Our method requires only a small number of sequences for effective pruning. To verify this point, we prune the Qwen3-30B-A3B-Thinking model at a 50%

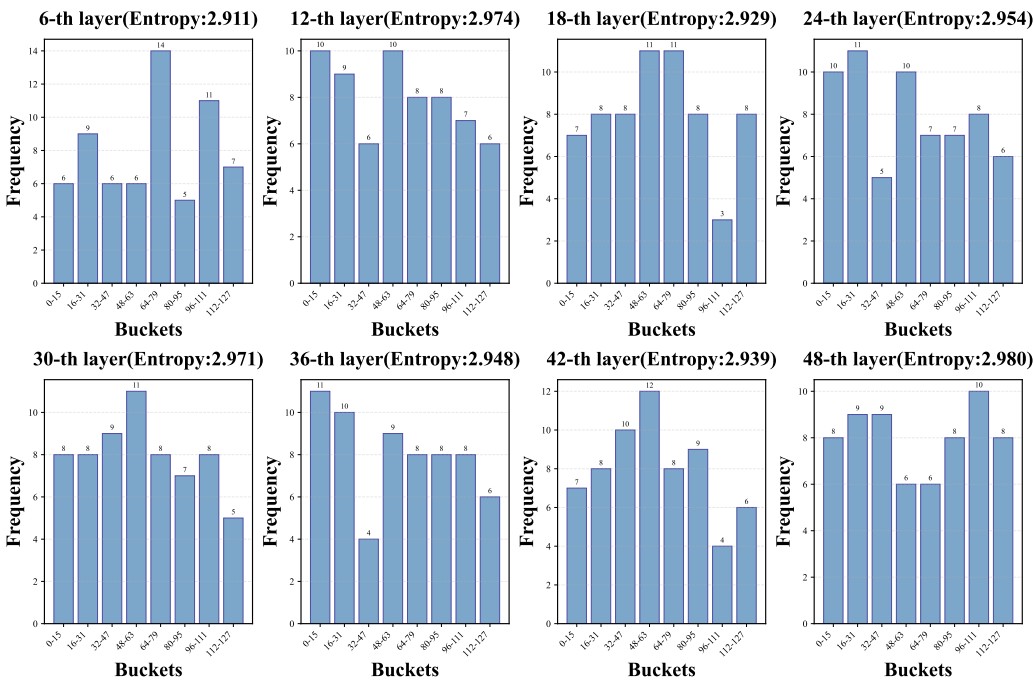

Figure 4: Expert Selection Distribution for Order-Based Grouping.

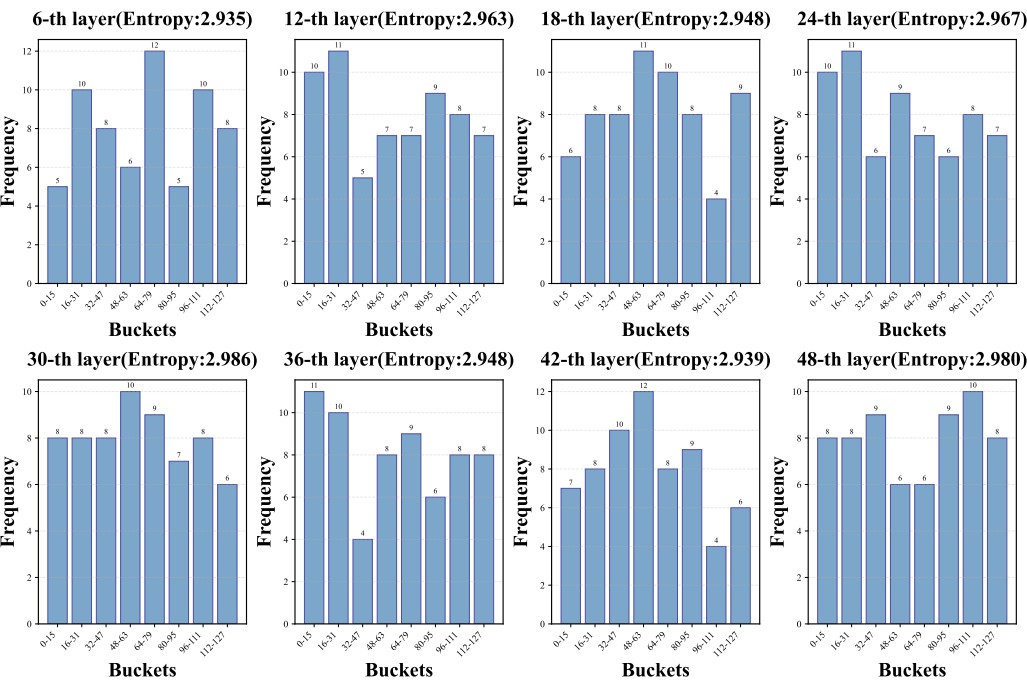

Figure 5: Expert Selection Distribution for Random Grouping.

pruning ratio using mixed datasets. As shown in Table 8, performance plateaus at 24 sequences (average 70.09%), with no significant improvement when increasing to 32 sequences (average 70.14%). This indicates that even with a small calibration set, our approach achieves stable pruning performance.

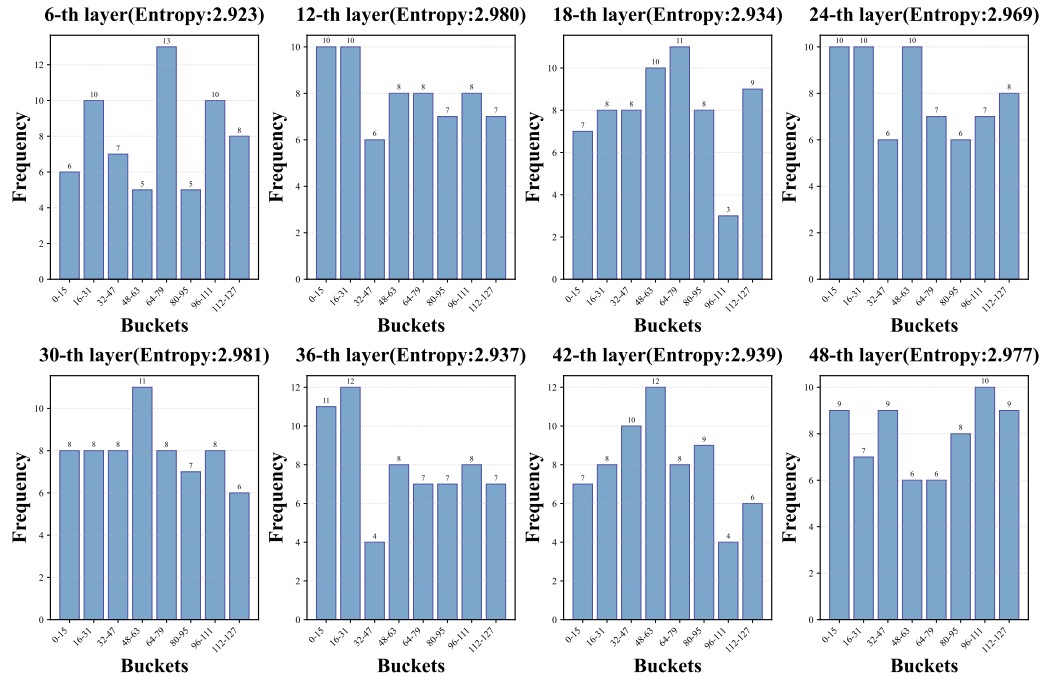

Figure 6: Expert Selection Distribution for Similary-based grouping.

Table 8: Performance with Different Sequences.

| Number of sequences | MMLU | Math500 | AIME25 | LCB | Average |
|---|---|---|---|---|---|
| 8 | 56.08 | 95.60 | 73.30 | 6.10 | 57.77 |
| 16 | 53.25 | 96.20 | 76.70 | 50.00 | 69.04 |
| 24 | 52.17 | 96.20 | 80.00 | 52.00 | 70.09 |
| 32 | 52.56 | 95.60 | 80.00 | 52.40 | 70.14 |

**Analysis of Different Pruning Ratios.** We compare our method with other approaches across varying pruning ratios. The results in Table 9 show that our method consistently outperforms the baselines, demonstrating its robustness with respect to the pruning ratio.

Table 9: Performance Comparison Across Pruning Ratios. The pruning ratio indicates the proportion of experts retained after pruning.

| Pruning ratios | Methods | MMLU | Math500 | AIME25 | LCB | Average |
|---|---|---|---|---|---|---|
| 100% | Original | 78.59 | 96.12 | 85.00 | 66.00 | 81.43 |
| 75% | Weights | 55.39 | 95.60 | 70.00 | 16.50 | 59.37 |
| | HC-SMoE | 65.77 | 91.80 | 70.00 | 66.50 | 73.52 |
| | Ours | 77.46 | 96.40 | 86.67 | 68.60 | 82.28 |
| 50% | Weights | 47.16 | 16.20 | 0.00 | 0.00 | 15.84 |
| | HC-SMoE | 43.54 | 37.40 | 16.70 | 29.60 | 31.81 |
| | Ours | 52.56 | 95.60 | 80.00 | 52.40 | 70.14 |
| 25% | Weights | 23.28 | 0.01 | 0.00 | 0.00 | 5.82 |
| | HC-SMoE | 22.95 | 2.80 | 0.00 | 0.00 | 6.44 |
| | Ours | 32.89 | 84.40 | 33.33 | 0.00 | 37.66 |

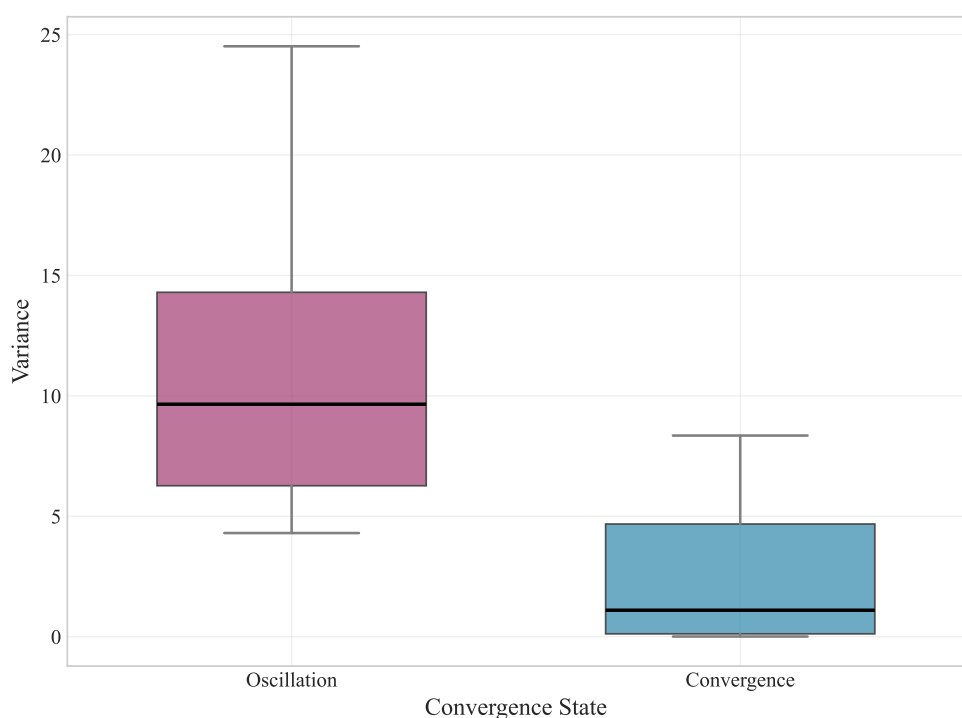

Figure 7: Statistical Distribution of Output Difference Variance.

**Reweighting Strategy Analysis.** We employ an exponential reweighting strategy ($\frac{\exp \boldsymbol{d}^t}{|\exp \boldsymbol{d}^t|_1}$) to update dataset sampling ratios, as detailed in Algorithm 2. For comparison, a linear reweighting strategy ($\frac{\boldsymbol{d}^t}{|\boldsymbol{d}^t|_1}$) is also evaluated. As presented in Table 10, the exponential strategy achieves a higher average score (70.14) than the linear strategy (62.82) within 3 iterations. The resulting dataset ratios are 2:1:1 for the exponential strategy and 10:3:3 for the linear strategy.

Table 10: Comparison with linear reweighting strategy.

| Reweighting strategy | Average | Dataset ratios | Max iterations |
|---|---|---|---|
| Exponential reweighting | 70.14 | 2:1:1 | 3 |
| Linear reweighting | 62.82 | 10:3:3 | 3 |

**Convergence Analysis of DCDM.** To verify the convergence of DCDM (Algorithm 2), we introduce a perturbation vector $\boldsymbol{\epsilon}$ into the update rule, defined as $\frac{\exp(\boldsymbol{\epsilon} \odot \boldsymbol{d}^t)}{|\exp(\boldsymbol{\epsilon} \odot \boldsymbol{d}^t)|_1}$. Each element of $\boldsymbol{\epsilon}$ is sampled from a uniform distribution over the interval $(0, 10]$. We then compute the variance of the output difference after applying this perturbation. A trial is labeled as convergence if it satisfies the stopping criterion in Algorithm 2; otherwise, it is labeled as oscillation. By visualizing the relationship between this variance and the convergence status, we observe that lower variance is associated with convergence, as shown in Figure 7.

In our experimental settings, the variance remains consistently below the mean value observed for converged cases, which explains the convergence behavior of Algorithm 2.

**Load-balancing Analysis.** We compare the expert activation distributions between the original Qwen3-30B-A3B-Thinking model (Figure 8) and the pruned model (Figure 9). The results indicate that the pruned model exhibits a flatter distribution. To further assess the load-balancing effect, we compute the Gini coefficients for both models, as shown in Table 7. A lower Gini coefficient reflects better load balance. The pruned model consistently achieves lower Gini coefficients across all layers, demonstrating improved load balancing.

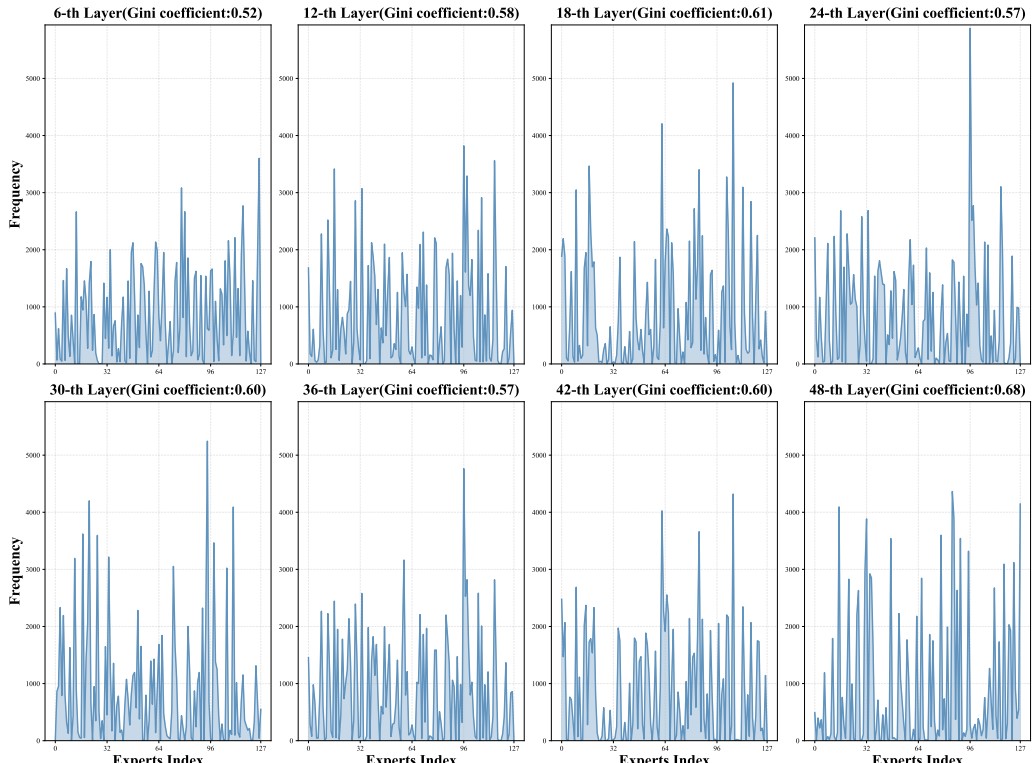

Figure 8: Expert activation frequency across layers in the original Qwen3-30B-A3B-Thinking model.

Furthermore, we evaluate the throughput of the original and pruned models on the C4 dataset using EvalScope (Team, 2024a) On a single A800 GPU. The pruned model achieved a throughput of 5384.07 tokens/s, compared to 4408.41 tokens/s for the original model. This result demonstrates enhanced inference efficiency without compromising routing quality.

**Analysis of Cross-Domain and Single-Domain Calibration.** From Table 2, we identify two fundamental principles governing the interaction between cross-domain mixing and single-domain calibration during pruning: (1) domain conflict leads to degradation, and (2) domain support leads to improvement. First, we observe that the Code domain (LCB) is the most vulnerable to cross-domain interference. Pruning with either C4 or OpenR1-Math introduces severe degradation, indicating strong domain conflict. In such cases, mixing data from conflicting domains offers no benefit and single-domain calibration remains the optimal choice. In contrast, the Math domain exhibits the opposite pattern. Code data introduces almost no harm to Math pruning, with only a very small decrease in accuracy (93.08 *vs.* 94.60 for Math500). This near-neutral interaction suggests that mixing compatible domains, *i.e.,* OpenR1-Math and rStar-Coder, can potentially yield better performance than single-domain calibration alone. These findings highlight that the effectiveness of cross-domain mixing is highly domain-dependent. However, cross-domain behavior is also influenced by additional factors, such as differences in domain sample size, data quality, and domain distribution, which still remain nontrivial to characterize. Understanding these interactions in a more principled way is an open question, and we leave a deeper investigation of data mixing strategies for future work.

**Post-training Analysis.** As shown in Figure 10, we fine-tuned the pruned Qwen3-30B-A3B-Thinking model using LoRA. Our experimental setup is consistent with the MoE example provided in the framework (Zhao et al., 2024). The loss curves indicate that our method consistently achieves lower loss values compared to other approaches, demonstrating the effectiveness of our pruning technique in preserving model capacity.

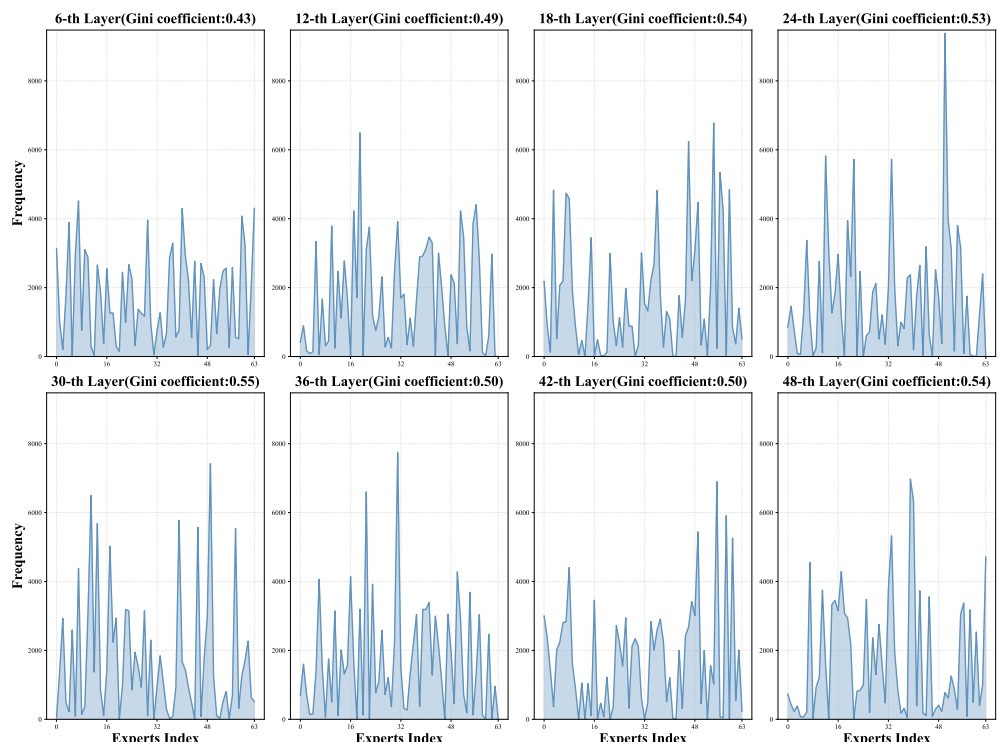

Figure 9: Expert activation frequency across layers after 50% pruning in Qwen3-30B-A3B-Thinking.

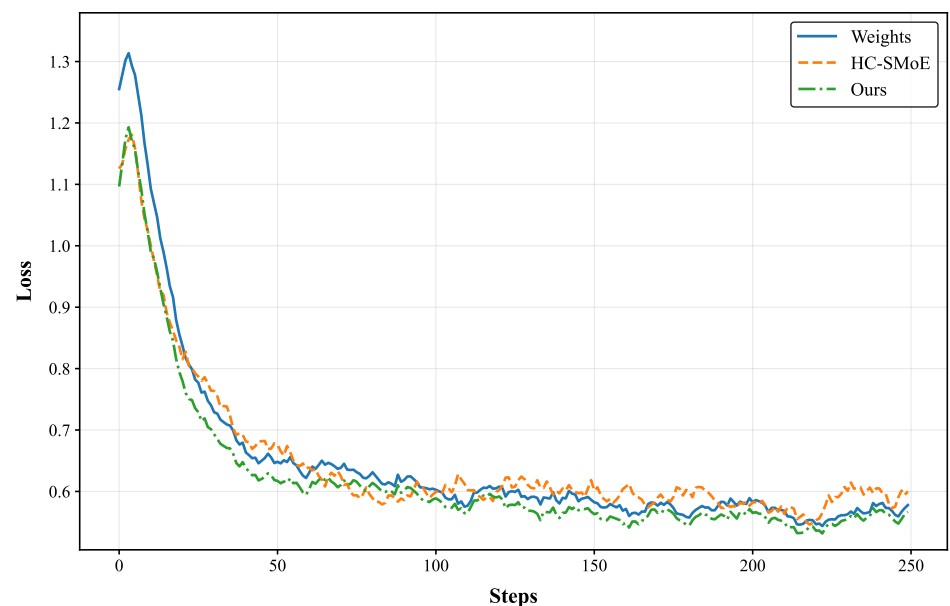

Figure 10: Loss Curves of Pruned Models after Fine-tuning. (Qwen3-30B-A3B-Thinking, 50% pruned pruning).

# D  ADDITIONAL RESULTS

**Evaluation on Additional Datasets.** To further validate the effectiveness of our method pruned on mixed datasets, we extend our evaluation to a broader range of benchmarks. As shown in Table 11,

our approach consistently outperforms the others. Specifically, it surpasses HC-SMoE by over 13% (58.69% vs. 45.24%).

Table 11: Performance comparison across diverse benchmarks on pruned mixed datasets (Qwen3-30B-A3B-Thinking, 50% pruned). H.S. is denoted as HC-SMoE. GPQA is denoted as GPQA_DIAMOD. H.E. is denoted as HumanEval+. AIME is denoted as AIME25. Math is denoted as Math500.

| Method | MMLU | GPQA | BBH | GSM8K | H.E.+ | MBPP+ | MBPP | Math | AIME | LCB | Avg |
|--------|------|------|------|-------|-------|-------|------|------|------|------|------|
| Original | 78.59 | 43.43 | 65.67 | 96.06 | 85.37 | 73.28 | 68.80 | 96.12 | 85.00 | 66.00 | 73.03 |
| Weights | 47.16 | 21.86 | 63.43 | 42.30 | 0.00 | 0.00 | 0.00 | 16.20 | 0.00 | 0.00 | 24.96 |
| H.S. | 43.54 | 22.22 | 62.42 | 69.67 | **51.22** | 40.21 | 27.40 | 37.40 | 16.70 | 29.60 | 45.24 |
| Ours | **52.56** | **32.83** | **66.66** | **93.93** | **51.22** | **63.23** | **50.40** | **95.60** | **80.00** | **52.40** | **58.69** |

**Comparison with Structural Pruning Baselines.** Structural pruning is a well-established technique in traditional dense models. In this section, we compare our method against three representative structural pruning baselines:

- **Weight importance baselines** (Filters'Importance, 2016): We compute expert importance by summing the Frobenius norm of each expert's parameters.
- **Hypernetwork-based baselines** (Gao et al., 2024): We adopt the same hypernetwork architecture followed with ReinMax, training parameters and objective function(p=0.5). The hypernetwork outputs determine which experts are selected in each MoE layer.
- **Element-wise importance baselines** (Ma et al., 2023): We compute element-wise importance scores for each expert and aggregate them via summation to obtain final expert importance.

As shown in Table 12, our method achieves superior performance, with an average score of 70.14 compared to 56.20 from the best baseline.

Table 12: Performance comparison with structural pruning baselines.

| Method | MMLU | Math500 | AIME25 | LCB | Average |
|--------|------|---------|--------|------|---------|
| Original | 78.59 | 96.12 | 85.00 | 66.00 | 81.43 |
| Weight importance | 23.76 | 9.20 | 6.70 | 0.00 | 9.92 |
| Hypernetwork-based | 52.01 | 89.20 | 40.00 | 18.30 | 49.88 |
| Element-wise importance | 43.58 | 89.20 | 63.30 | 28.70 | 56.20 |
| Ours | **52.56** | **95.60** | **80.00** | **52.40** | **70.14** |

