# OpenReview forum: "Compressing Large MoE Models via Efficient Pruning and Data-Aware Calibration"
_ICLR.cc/2026/Conference — Submitted to ICLR 2026_

### Official Review · Reviewer_yUmm · 2025-10-17

**Soundness:** 2
**Presentation:** 2
**Contribution:** 2
**Rating:** 6
**Confidence:** 4

**Summary:**

This paper addresses expert pruning in large Mixture-of-Experts models through two main contributions: a coarse-to-fine selection strategy reducing complexity from O(C(2n,n)) to O(n^1.5), and a dynamic calibration dataset mixing (DCDM) approach for cross-domain generalization. Theorem 4.1 bounds global discrepancy by layer-wise error accumulation. Experiments on DeepSeek-R1 (671B parameters) and Qwen3-30B retain 98.9% performance while pruning 50% of experts. While computationally efficient and theoretically grounded, the work requires deeper investigation of grouping strategies and domain-specific trade-offs.

**Strengths:**

1. **The complexity reduction is substantial and well-demonstrated.** The coarse-to-fine strategy achieves O(n^1.5) versus O(n^2) for naive greedy search, validated through concrete comparison: 2.06×10⁵ evaluations versus 58×C(256,128) for baseline methods on DeepSeek-R1. This represents meaningful practical improvement for large-scale deployment.

2. **Theorem 4.1 provides solid theoretical grounding.** The layer-wise bound justifies greedy optimization, with Figure 2 confirming tight empirical bounds (Lipschitz constants <0.5, discrepancies ≤1.4×10⁻⁷). This connection between theory and practice strengthens the methodological foundation.

3. **DCDM addresses genuine cross-domain degradation.** Table 2 clearly demonstrates the problem: C4 calibration yields 59.91 on MMLU but 0.00 on LCB. The adaptive reweighting provides a principled solution, achieving 70.14 average versus 68.07 for best single-domain approach.

**Weaknesses:**

1. **The grouping strategy lacks empirical validation.** Sequential partitioning into groups of size S is used without justification. Given the greedy nature of coarse-to-fine selection, initial grouping could significantly impact results if similar experts cluster together. The paper claims iterative regrouping mitigates this but provides no evidence. Critical missing experiments: performance variance across random groupings, comparison with similarity-based grouping, and analysis of whether optimal experts distribute uniformly or cluster in sequential ordering.

2. **Domain-specific performance trade-offs are inadequately evaluated.** Table 2 shows DCDM achieves higher average scores, but the fundamental question remains unaddressed: does cross-domain mixing sacrifice peak single-domain performance? The shown comparisons (DCDM vs. mismatched calibration) differ in both method and calibration domain, confounding interpretation. When practitioners require optimal performance on specific domains (e.g., deploying solely for mathematical reasoning), does DCDM match or underperform domain-specific calibration with matched data? Table 2 suggests potential degradation (DCDM: 95.60 on Math500 vs. OpenR1-Math: 94.60), but lacks direct controlled comparison and discussion of deployment scenarios favoring specialization over generalization.

**Questions:**

1. **Can you quantify the impact of initial grouping on final performance?** Report variance across multiple random groupings, compare sequential versus similarity-based partitioning, and analyze whether high-quality experts distribute uniformly in your ordering scheme.

2. **Does DCDM compromise peak single-domain performance?** Provide direct comparison: DCDM versus domain-matched single-domain calibration, evaluated on that domain's benchmarks. When should practitioners choose DCDM over specialized pruning?

3. **What justifies the exponential reweighting in DCDM?** Compare against linear or other update rules. Analyze convergence: typical iteration counts, stability of final weights, and behavior when stopping criterion isn't met.

---

> ### Author Response · Authors · 2025-11-23
>
> Thank you for the helpful suggestions and detailed comments. We outline our responses and clarifications below. All corresponding updates—including supplementary tables, additional analyses, extended discussions, and any necessary explanations—will be incorporated into the revised manuscript.
>
> - **W1&Q1**: Missing comparisons with random and similarity-based grouping.
>
> We address this concern from two perspectives: a performance comparison of grouping strategies and an analysis of expert distributions.
>
> （1）**Performance comparision.**
>
> We evaluated three expert grouping strategies after pruning 50% of experts in Qwen3-30B-A3B-Thining.
>
> `Random grouping`: experts are shuffle randomly before group.
>
> `Similarity-based grouping`: experts are grouped by greedily selecting the most similar candidates for each group.
>
> `Order-based grouping`: experts are grouped by their original indices.
>
> As shown in Table 1, all grouping strategies preserve competitive performance, with order-based grouping slightly outperforming the others.
>
> Table 1. Performance under Different Grouping Strategies.
>
> | Method         | MMLU   | Math500 | AIME25 | LCB   | Average |
> |----------------|--------|---------|--------|--------|----------|
> | Original       | 78.59 | 96.12  | 85.00  | 66.00 | 81.43    |
> | Random         | **52.59** | 92.20  | 73.30  | 47.10 | 66.30    |
> | Similarity-based | 52.56 | 95.20  | **80.00** | 51.50 | 69.82    |
> | Order-based    | 52.56 | **95.60** | **80.00** | **52.40** | **70.14** |
> |
>
> (2) **Analysis of router behavior.**
>
> We further analyzed router behavior by examining the distribution of selected experts using 8 buckets (bucket size = 16). As illustrated in Figures 4–6  (see Appendix C) of our manuscript, the distributions of selected experts appear highly uniform across all grouping strategies. This observation is then quantified by the high entropy values (all close to the maximum of 3), confirming near-uniform expert selection (Table 2). These results collectively suggest that the performance of our method is robust to the choice of grouping strategy. This suggests that the performance of our method is not strongly influenced by the choice of grouping strategy.
>
> Table 2. Entropy of expert selection after 50% pruning
>
> | Grouping Method | 6 | 12 | 18 | 24 | 30 | 36 | 42 | 48 |
> |-----------------|---|---|---|---|---|---|---|---|
> | Order-based | 2.91 | 2.97 | 2.93 | 2.95 | 2.97 | 2.95 | 2.94 | 2.98 |
> | Random | 2.94 | 2.96 | 2.95 | 2.98 | 2.99 | 2.95 | 2.94 | 2.98 |
> | Similarity-based | 2.92 | 2.98 | 2.93 | 2.97 | 2.98 | 2.94 | 2.94 | 2.98 |
> |
>
> - **W2&Q2**: Lack of analysis on the trade-off between cross-domain mixing and domain-specific peak performance
>
> We appreciate the reviewer’s concern regarding potential trade-offs between cross-domain mixing and peak single-domain performance. Our perspective is that whether cross-domain mixing helps or hurts a domain is not arbitrary, but follows a simple and interpretable pattern: if a model pruned using domain A performs significantly worse when evaluated on domain B, then mixing A and B is unlikely to help B; conversely, if pruning on A does not degrade B—or even improves it—then including A in the mixture is likely to further benefit B. This heuristic aligns with the empirical behavior we observe across tasks: domains whose calibration data are mutually supportive tend to benefit from mixing, while domains that interfere with each other do not. We will clarify this principle in the final version.
>
> - **Q3**: Comparisions with linear rules
>
> Thank you for the question. In our experiments on Qwen3-30B-A3B-Thinking, we compared the linear update rule with our exponential reweighting approach and found that both methods converge to a stable domain ratio within only a few iterations (no more than three in all cases). However, they converge to different final ratios: the linear rule stabilizes around 10:3:3, whereas the exponential strategy converges to 2:1:1. Both linear and exponential schemes exhibit stable behavior under our setup; if the stopping criterion is not triggered earlier, the updates simply continue until reaching the preset maximum iteration count.
>
> In addition, we compare our exponential reweighting strategy with the linear strategy across four datasets—MMLU, Math500, AIME25, and LCB. The exponential method achieves a higher average score than the linear one (70.14 vs. 62.82), further demonstrating the effectiveness of our approach.

---

> > ### Comment · Reviewer_yUmm · 2025-11-24
> >
> > The authors' rebuttal has partially addressed my concerns. However, the authors have not provided a revised PDF manuscript that comprehensively incorporates the improvements suggested by reviewers. I will maintain my current score and hope that the authors can revise the manuscript as soon as possible.

---

> > > ### Author Response · Authors · 2025-11-24
> > > **Revised Submission and Correspondence to Updated Sections**
> > >
> > > We thank the reviewer for the valuable comments. We have submitted a revised version of the paper, and below we summarize how each response corresponds to the updated sections in the revision. If there are any remaining concerns, we are fully open to further discussion and happy to clarify anything that may still be unclear.
> > >
> > > | Reviewer Issue | Question ID | Corresponding Appendix Section |
> > > |----------------|-------------|--------------------------------|
> > > |  Performance comparison concerns | W1 & Q1 | **Appendix C: Grouping Strategies Analysis** |
> > > | Router behavior and load distribution | W1 & Q1  | **Appendix C: Distribution of Pruned Experts Analysis** |
> > > | Cross-domain vs. single-domain pruning | W2 & Q2 | **Appendix C: Analysis of Cross-Domain and Single-Domain Calibration** |
> > > | Reweighting / mixing strategy justification | Q3 | **Appendix C: Reweighting Strategy Analysis** |
> > > |

---

> > > > ### Comment · Reviewer_yUmm · 2025-11-25
> > > >
> > > > The authors addressed my concerns well. The added experiments and explanations strengthen the paper. I'm maintaining my positive score.

---

> ### Author Response · Authors · 2025-11-26
> **Thank you for the constructive feedback**
>
> We are very glad that our response has fully addressed your concern. We would be truly grateful if you feel it is appropriate to reconsider your rating. Thank you again for your time and constructive feedback.

---

### Official Review · Reviewer_vzcY · 2025-10-19

**Soundness:** 3
**Presentation:** 2
**Contribution:** 2
**Rating:** 4
**Confidence:** 4

**Summary:**

This paper introduces a pruning framework designed to tackle memory overhead challenges when deploying large Mixture-of-Experts (MoE) models. The research focuses on addressing redundancy in large MoE models, where high memory costs from inactive experts create deployment barriers. Current reconstruction-loss-based pruning strategies struggle with two key issues: computationally prohibitive exhaustive searches for evaluating expert combinations and reliance on fixed calibration datasets that cannot preserve cross-domain performance. To overcome these limitations, the authors develop a framework featuring two key contributions: (1) a Coarse-to-Fine Expert Selection strategy that employs hierarchical greedy search with group-level scoring to reduce computational complexity and (2) a Dynamic Calibration Dataset Mixing strategy that adaptively reweights domain sampling based on pruning-induced discrepancies to improve generalization across domains.

**Strengths:**

- The paper provides theoretical justification for its layer-wise pruning approach, proving that the global output difference can be bounded by the accumulation of local layer-wise discrepancy, which makes the layer-wise strategy more convincing and well-grounded.
- The paper validates its approach on a large-scale MoE model (DeepSeek-R1), which is a true state-of-the-art open-source model, making the conclusions more credible and practically relevant.

**Weaknesses:**

- The paper only provides complete baseline comparisons on Qwen3-30B-A3B-Thinking (two baselines across four datasets), while DeepSeek-R1 lacks proper baseline comparisons and only reports results on two datasets. The evaluation suffers from insufficient SOTA baselines, limited dataset coverage (only four or two datasets), and lacks evaluation on important benchmarks (e.g., GPQA).
- The evaluation lacks testing across different model scales and families, which limits the validation of the method's generalizability and robustness across diverse architectures.
- The paper only tests at a single 50% sparsity ratio without exploring other compression levels, limiting the understanding of the method's performance across different pruning intensities.
- The main approach using layer-wise reconstruction-loss-based compression is similar to prior works (e.g., [1]), making the novelty incremental rather than substantial.

[1] Not All Experts are Equal: Efficient Expert Pruning and Skipping for Mixture-of-Experts Large Language Models

**Questions:**

1. How does the method perform with comprehensive baseline comparisons against multiple SOTA methods and on more diverse benchmarks?
2. How does the method perform across different model scales and families?
3. What is the performance of the proposed method at different sparsity ratios?

---

> ### Author Response · Authors · 2025-11-23
> **Rebuttal for weakness and questions W1&W2&Q1&Q2**
>
> Thank you for the insightful feedback. We provide our detailed responses and clarifications below. All relevant additions—such as updated tables, further analyses, extended discussions, and necessary explanations—will be fully reflected in the revised manuscript.
>
> - **W1&W2&Q1&Q2**: Insufficient baseline methods, datasets and model variety
>
> The concerns can be summarized into the following four key points: 1) DeepSeek-R1 lacks proper baselines and datasets. 2) Insufficient SOTA baselines. 3)Limited datasets. 4) Narrow Model Diversity. We have conducted extensive additional experiments to address each of these points as follows:
>
> (1) **We have added the weights method as an additional baseline for DeepSeek-R1**. As shown in Table 1, our method achieves substantially better performance across all metrics compared to this baseline.
>
> Table 1. Comparsion with weights methods.
>
> | Method  | Math500 | AIME25 | LCB   | Average |
> |---------|---------|--------|-------|---------|
> | Weights | 63.6    | 32.08  | 52.69 | 49.46   |
> | Ours    | **95.2**    | **62.5**   | **60.22** | **72.64**   |
> |
>
> (2) **We have included more comparisions with SOTA baselines as the reviewer suggested.**
>
> Thank you for your comment regarding baseline selection. We selected HC-SMoE as it was the most recent open-source MoE pruning method available prior to our ICLR submission. We believe this represents a sufficiently strong and current baseline for comparison.
>
> In addition to the original weight-based and HC-SMoE baselines, we have incorporated the additional baseline methods suggested by Reviewer wtZA, including weight-importance, hypernetwork-based, and element-wise importance pruning, as well as the Data-Centric and Model-Centric Expert-guided methods introduced in [a]. As shown in Table 2, our method consistently outperforms all of them.  The corresponding experimental results are provided in Appendix D: Comparison with Structural Pruning Baselines.
>
> To ensure a fair and direct comparison with the Data-Centric and Model-Centric approaches, we closely followed the experimental setup described in [a], evaluating on the DeepSeek-16B model using the MMLU, BoolQ, OpenBookQA, and RTE benchmarks. These results are presented in Table 4 of Section 4 (Different Model Scales), where our method again demonstrates superior performance over all baselines.
>
> Table 2.  Comparison with Structural Pruning Baselines.
>
> | Method            | MMLU   | Math500 | AIME25 | LCB    | Average |
> |-------------------|--------|---------|--------|--------|----------|
> | Original          | 78.59  | 96.12   | 85.00  | 66.00  | 81.43    |
> | Weight Importance | 23.76  | 9.20    | 6.70   | 0.00   | 9.92     |
> | Hypernetwork-based | 52.01| 89.20   | 40.00  | 18.30| 49.88   |
> | Element-wise importance | 43.58| 89.20   | 63.30  | 28.70| 56.20   |
> | Ours              | **52.56** | **95.60** | **80.00** | **52.40** | **70.14** |
> |
>
> [a] Zhang, Z., Liu, X., Cheng, H., Xu, C., & Gao, J. Diversifying the expert knowledge for task-agnostic pruning in sparse mixture-of-experts. Findings of ACL 2025.
>
> (3) **Limited datasets**
>
> We have expanded our experiments to include a wider range of benchmarks—covering mathematics, code, general knowledge, and reasoning. As shown in Table 3,  our method consistently outperforms other baselines across the following tasks.
>
> Table 3. Performance comparison across diverse benchmarks on mixed datasets (Qwen3-30B-A3B-Thinking, 50% pruned). GPQA is denoted as GPQA_DIAMOD.
>
> | Dataset   | MMLU   | GPQA   | BBH   | GSM8K | HumanEval_Plus | MBPP_Plus | MBPP  | Average |
> |-----------|--------|--------|-------|--------|-----------------|------------|--------|----------|
> | Original  | 78.59  | 43.43  | 65.67 | 96.06 | 85.37          | 73.28      | 68.80 | 73.03   |
> | weighted  | 47.16  | 21.86  | 63.43 | 42.30 | 0.00           | 0.00       | 0.00  | 24.96   |
> | HC-SMoE   | 43.54  | 22.22  | 62.42 | 69.67 | 51.22          | 40.21      | 27.40 | 45.24   |
> | Ours      | **52.56** | **32.83** | **66.66** | **93.93** | **51.22** | **63.23** | **50.40** | **58.69** |
> |
>
> (4) **Different model scales.**
>
> We have evaluated our method across models of varying scales, including DeepSeek-MoE-16B (16B), Qwen3-30B-A3B-Thinking (30B), and DeepSeek-R1 (671B), covering a parameter range from 16B to 671B.
>
> Table 4. Comparsion with Data-Centric Expert-guided and Model-Centric Expert-guided methods. Baseline results are evaluated from Figure E4 in [a].
>
> | Method                     | MMLU  | BoolQ | OpenBook QA | RTE   | Average |
> |----------------------------|-------|--------|--------------|--------|----------|
> | Original                   | 44.00 | 77.50 | 33.80        | 66.00 | 55.33    |
> | Data-Centric Expert-guided | 36.50 | 71.50 | 28.00        | 69.00 | 51.25    |
> | Model-Centric Expert-guided| 32.50 | 71.50 | **29.50**    | 63.00 | 49.13    |
> | Ours                       | **40.14** | **77.00** | 29.40 | **66.43** | **53.24** |
> |

---

> ### Author Response · Authors · 2025-11-23
> **Rebuttal for weakness 3,4 and the remaining questions**
>
> - **W3&Q3**: Lack of evaluation across multiple sparsity ratios
>
> Thank you for the suggestion. We have evaluated the robustness of our method by testing different pruning ratios (25%, 50%, 75%) on the Qwen3-30B-A3B-Thinking model using the mixed dataset.
>
> As shown in the following table , the model maintains performance at 75% pruning, shows a moderate decline at 50%, and experiences a significant drop at 25% pruning. This trend demonstrates that performance remains viable at 75% and 50% ratios but severely degrades under more aggressive pruning.
>
> Table 5. Performance under different pruning ratios.
>
> | Sparsity | MMLU  | Math500 | AIME25 | LCB   | Average |
> |----------|--------|---------|--------|--------|----------|
> | Original | 78.59 | 96.12  | 85.00  | 66.00 | 81.43    |
> | 75%      | 77.46 | 96.40  | 86.67  | 68.60 | 82.28    |
> | 50%      | 52.56 | 95.60  | 80.00  | 52.40 | 70.14    |
> | 25%      | 32.89 | 84.40  | 33.33  | 0.00  | 37.66    |
> |
>
> - **W4**: Comparision with  reconstruction-loss-based baseline
>
> Directly using the reconstruction-loss approach from [1] on a model the scale of DeepSeek-R1 is not feasible in practice, as its computational cost grows exponentially—an issue consistently acknowledged in multiple prior studies[2]. Thus, our contribution goes beyond reusing the reconstruction-loss idea; it lies in making this class of pruning methods practical and theoretically justified for large-scale MoE models.
>
> **First**, on the algorithmic side, we introduce a coarse-to-fine approximation that dramatically reduces the search complexity—from exponential to polynomial—making reconstruction-based pruning computationally feasible even for models with hundreds of experts per layer. **Second**, on the theoretical side, we prove that the approximation-induced discrepancy is provably controlled through an explicit upper bound, ensuring that the reduced computation does not lead to unbounded error. These two components jointly enable reconstruction-based MoE pruning at scales that prior approaches cannot support.
>
> - **Q1**: Insufficient baseline methods and datasets
> See the response 2) and 3) for Weakness 1.
>
> [1] Xudong Lu, Qi Liu, Yuhui Xu, Aojun Zhou, Siyuan Huang, Bo Zhang, Junchi Yan, Hongsheng Li. Not All Experts are Equal: Efficient Expert Pruning and Skipping for Mixture-of-Experts Large Language Models. ACL 2024.
>
> [2] I-Chun Chen, Hsu-Shen Liu, Wei-Fang Sun, Chen-Hao Chao, Yen-Chang Hsu, Chun-Yi Lee. Retraining-Free Merging of Sparse MoE via Hierarchical Clustering. ICML 2025.

---

> ### Author Response · Authors · 2025-11-26
> **Gentle Reminder**
>
> This is a gentle reminder regarding the review for our submission. If there are any points in our paper that remain unclear or if further clarification from our side would help you complete the review, please feel free to let us know. We would be more than happy to assist in making our work easier to evaluate.

---

### Official Review · Reviewer_RQgE · 2025-10-28

**Soundness:** 3
**Presentation:** 2
**Contribution:** 3
**Rating:** 4
**Confidence:** 4

**Summary:**

This paper proposes a pruning framework for ultra-large Mixture-of-Experts LLMs that combines (i) a coarse-to-fine, layer-wise greedy expert selection based on output reconstruction discrepancy and (ii) a Dynamic Calibration Dataset Mixing strategy that adapts the calibration mixture across domains during pruning.

**Strengths:**

[1] The coarse-to-fine expert selection within a layer is an interesting idea that trades exhaustive evaluation for a two-stage screening (group-level then expert-level) guided by a reconstruction loss rather than router statistics.
[2] DCDM brings ideas from data-mixture optimization into post-training compression, adaptively up-weighting domains where pruning induces larger output discrepancy, which is under-explored in MoE pruning.
[3] The paper is generally well structured.

**Weaknesses:**

[1] The selection uses an L2 reconstruction discrepancy on a small calibration set (32 sequences). While DCDM adapts the mix, the overall budget is tiny relative to model scale and task diversity.
[2] The theoretical bound assumes layer-wise independence in the propagation of discrepancies, effectively treating each layer’s pruning error as separable and additive.
[3] Limited analysis of router behavior and load balancing post-pruning.  Pruning half the experts may shift router distributions, hit the shared expert more heavily, or alter token load balance.
[4] Limited experiments. The paper only did experiments on Qwen3-30B-A3B-Thinking and DeepSeek-R1. Only a setting of the experiment (~50%) is conducted, and the baseline results are only included in the experiment of the Qwen model.

**Questions:**

See weakness.
(1) It is better to vary the pruning ratio to see the performance trend for different methods to check the algorithm's robustness.
(2) It is better to include the efficiency comparison between pre- and post-pruned MoE models.
(3) It seems that the activation-based pruning method, like [a] only requires O(N), while the authors did not consider it in the discussion.


[a] Zhang, Z., Liu, X., Cheng, H., Xu, C., & Gao, J. Diversifying the expert knowledge for task-agnostic pruning in sparse mixture-of-experts. Findings of ACL 2025.

---

> ### Author Response · Authors · 2025-11-23
> **Rebuttal for weakness and questions W1, W2, W3&Q2**
>
> Thank you for the valuable suggestions and constructive comments. We address each point in detail below and provide the corresponding clarifications. All related updates—including additional tables, analyses, discussions, and explanations—will be incorporated into the revised manuscript.
>
> - **W1**: Tiny calibration budget vs. model scale
>
> Thank you for raising this point. As shown in Table 1,  we have evaluated the impact of calibration set size by testing different numbers of sequences (from 8 to 32) when pruning the Qwen3-30B-A3B-Thinking model on mixed datasets. The results show that performance plateaus at 24 sequences (average 70.09%), with no significant improvement when increasing to 32 sequences (average 70.14%). This indicates that while the calibration set is small, it is sufficient to achieve stable pruning performance in our setup.
>
> Table 1. Performance with Different Sequences.
>
> | Number of sequences | MMLU  | Math500 | AIME25 | LCB   | Average |
> |---------------------|-------|---------|--------|-------|---------|
> | 8                   | 56.08 | 95.60   | 73.30  | 6.10  | 57.77   |
> | 16                  | 53.25 | 96.20   | 76.70  | 50.00 | 69.04   |
> | 24                  | 52.17 | 96.20   | 80.00  | 52.00 | 70.09   |
> | 32                  | 52.56 | 95.60   | 80.00  | 52.40 | 70.14   |
> |
>
> - **W2**: Theoretical Layer-wise Independence Assumption
>
> Thank you for your comment. We would like to clarify that our theoretical proof does not rely on any layer-wise independence assumption. The core of the proof is based on the Lipschitz continuity of each layer and the application of the triangle inequality, which bridges the error bounds between global pruning and layer-wise pruning methods. The complete derivation is provided in Appendix B.
>
> This theorem shows that the discrepancy between the original and pruned model outputs can be bounded by the accumulation of layer-wise differences, thereby providing a theoretical justification for the effectiveness of layer-wise pruning in practice.
>
> - **W3&Q2**: Insufficient router/load-balancing analysis/Throughput
>
> Thank you for raising this important question regarding router behavior after pruning. As shown in Figures 7-8  (see Appendix C) of our manuscript, the expert activation distributions demonstrate that the pruned model exhibits a more balanced activation pattern compared to the original model.
>
> This improved balance is quantitatively confirmed by the consistently lower Gini coefficients across all layers in the pruned model (Table 2), indicating a more uniform distribution of expert utilization.
>
> Table 2. Gini coefficients (original vs. pruned).
>
> | Model    | 6    | 12   | 18   | 24   | 30   | 36   | 42   | 48   |
> |----------|------|------|------|------|------|------|------|------|
> | Original | 0.52 | 0.58 | 0.61 | 0.57 | 0.60 | 0.57 | 0.60 | 0.68 |
> | Pruned   | 0.43 | 0.49 | 0.54 | 0.53 | 0.55 | 0.50 | 0.50 | 0.54 |
> |
>
> Regarding the shared experts, since they remain always-activated as in the original model, their routing frequency remains unaffected by the pruning process.
>
> Additionally, the pruned model achieves a throughput of 5384.07 tokens/s compared to 4408.41 tokens/s for the original model, demonstrating enhanced inference efficiency while maintaining balanced routing.

---

> ### Author Response · Authors · 2025-11-23
> **Rebuttal for weakness 4 and remaining questions**
>
> - **W4&Q1&Q3**: Limited model variety, sparsity ratios, and pruning baselines
>
> Thank you for these constructive comments regarding the experimental scope. We have conducted additional experiments to address the three key points raised, as detailed below.
>
> (1) **Limited model variety**:
> We have evaluated a new model DeepSeek-16B model with 48 experts pruned on the C4 dataset. The results show that our method outperforms the methods in [a], which are summarized in Table 3:
>
> Table 3. Comparsion with Data-Centric Expert-guided and Model-Centric Expert-guided methods. Baseline results are evaluated from Figure E4 in [a]
>
> | Method                      | MMLU  | BoolQ | OpenBookQA | RTE   | Average |
> |-----------------------------|-------|-------|------------|-------|---------|
> | Original                    | 44.00 | 77.50 | 33.80      | 66.00 | 55.33   |
> | Data-Centric Expert-guided* | 36.50 | 71.50 | 28.00      | 69.00 | 51.25   |
> | Model-Centric Expert-guided*| 32.50 | 71.50 | **29.50**      | 63.00 | 49.13   |
> | Ours                        | **40.14** | **77.00** | 29.40      | **66.43** | **53.24**  |
> |
>
> (2) **Pruning ratios**:
>
> We have evaluated the robustness of our method by testing different pruning ratios (25%, 50%, 75%) on the Qwen3-30B-A3B-Thinking model using the mixed dataset.
>
> As shown in Table 4,  our method consistently outperforms the baselines across all pruning ratios, demonstrating its robustness with respect to the pruning ratio. These results are provided in Appendix C: Analysis of Different Pruning Ratios.
>
> **Table 4. Performance Comparison Across Pruning Ratios. The pruning ratio indicates the proportion of experts retained after pruning.**
>
> | Pruning ratios | Methods | MMLU | Math500 | AIME25 | LCB | Average |
> | :--- | :--- | :---: | :---: | :---: | :---: | :---: |
> | 100% | Original | 78.59 | 96.12 | 85.00 | 66.00 | 81.43 |
> | **75%** | **Weights** | 55.39 | 95.60 | 70.00 | 16.50 | 59.37 |
> | | **HC-SMoE** | 65.77 | 91.80 | 70.00 | 66.50 | 73.52 |
> | | **Ours** | 77.46 | 96.40 | 86.67 | 68.60 | 82.28 |
> | **50%** | **Weights** | 47.16 | 16.20 | 0.00 | 0.00 | 15.84 |
> | | **HC-SMoE** | 43.54 | 37.40 | 16.70 | 29.60 | 31.81 |
> | | **Ours** | 52.56 | 95.60 | 80.00 | 52.40 | 70.14 |
> | **25%** | **Weights** | 23.28 | 0.01 | 0.00 | 0.00 | 5.82 |
> | | **HC-SMoE** | 22.95 | 2.80 | 0.00 | 0.00 | 6.44 |
> | | **Ours** | 32.89 | 84.40 | 33.33 | 0.00 | 37.66 |
>
> (3) **Enhanced Baselines for DeepSeek-R1**:
>
> We have added the weights method as an additional baseline for DeepSeek-R1. As shown in Table 5, our method achieves substantially better performance across all metrics compared to this baseline.
>
> Table 5. Comparsion with weights methods.
>
> | Method  | Math500 | AIME25 | LCB   | Average |
> |---------|---------|--------|-------|---------|
> | Weights | 63.6    | 32.08  | 52.69 | 49.46   |
> | Ours    | **95.2**    | **62.5**   | **60.22** | **72.64**   |
> |
>
> These additional experiments demonstrate the robustness and generalizability of our method across different models, pruning ratios, and baseline comparisons.
>
> - **Q3**
>
> We would like to clarify that the time complexity of Algorithm 1 in [a] is O(N^2), not O(N), as evidenced by the nested loops in Lines 10–12 on page 101. To compare with the prior work[a], we  include a comparative evaluation in Table 3 (see W4&Q1&Q3). From the table, we can observe that  our method demonstrates superior performance over the method from [a].
>
> [a] Zhang, Z., Liu, X., Cheng, H., Xu, C., & Gao, J. Diversifying the expert knowledge for task-agnostic pruning in sparse mixture-of-experts. Findings of ACL 2025.

---

> ### Author Response · Authors · 2025-11-26
> **Gentle Reminder**
>
> This is a gentle reminder regarding the review for our submission. If there are any points in our paper that remain unclear or if further clarification from our side would help you complete the review, please feel free to let us know. We would be more than happy to assist in making our work easier to evaluate.

---

### Official Review · Reviewer_wtZA · 2025-10-29

**Soundness:** 2
**Presentation:** 3
**Contribution:** 2
**Rating:** 6
**Confidence:** 4

**Summary:**

The paper proposes a pruning framework for large MoE LMs with two main pieces: 1- Coarse-to-Fine Expert Selection that turns reconstruction-loss–based expert selection from exponential to polynomial time. 2- Dynamic Calibration Dataset Mixing (DCDM) that reweights calibration domains during pruning based on the observed output discrepancy between the original and pruned models.
Empirically, pruning 50% of routed experts on Qwen3-30B-A3B-Thinking retains ~86% of the original average score and outperforms clustering- and weight/activation-based baselines. On DeepSeek-R1, pruned models retain ~98.9% of the original across AIME25 and LCB. The paper also studies time vs. number of experts and group size, matching the polynomial analysis.

**Strengths:**

1- Scalable pruning algorithm for MoEs: Clear coarse-to-fine design with a measured reduction from exponential to $O(n^{1.5})$ behavior in practice.

2- Theoretical justification: A layer-wise pruning bound that justifies optimizing per-layer reconstruction instead of full-model comparison.

3- Data-aware calibration: DCDM adapts the calibration mix using discrepancy feedback and improves cross-domain retention over a fixed recipe.

4- Strong empirical results at scale: Competitive retention on Qwen3-30B-A3B-Thinking and ~99% retention on DeepSeek-R1.

5- Good ablations and time/cost analysis.

**Weaknesses:**

1- Baseline Coverage: Though there are clustering and simple weight/activate heuristic baselines are considered, the paper can be improved by including a simple hypernetwork-based baselines that predicts the retained experts based on the calibration dataset. Also, structural pruning baselines for proppsed for dense models are missing (weight importance can be aggregated for experts to have a single metric for each).

2- Incomplete related work: This paper should discuss layer-wise often greedy pruning algorithms proposed for neural networks, e.g. [1], [2], etc..

3- Limited task coverage:  Calibration is demonstrated on a small set of domains (math, code, general knowledge). The claim of broad cross-domain robustness would be stronger with more tasks. Qwen-3 is evaluate on MMLU, GPQA (variants), BBH, GSM8K, EvalPlus, MBPP, etc.

4- Limited novelty: Reconstruction loss metric is not new. The bound relies on standard Lipschitz-style accumulation, which is conceptually incremental even if useful for MoE. Data-mixture scheduling echoes prior work on mixture reweighting (e.g. DoReMi-style ideas as the paper mentions).

5- Missing inference latency /throughput values after pruning.

[1] ThiNet: A Filter Level Pruning Method for Deep Neural Network Compression, Luo et. al, 2017

[2] Efficient DNN Neuron Pruning by Minimizing Layer-wise Nonlinear Reconstruction Error∗, Jiang et. al , 2017

**Questions:**

1- How are experts grouped in algorithm 1? I can't seem to find this information in the paper.

2- Why are other pruning criteria not reported for the mixed row in table 2?

3- Can you provide some insights on how often the weight update oscillates among domains.

4- How Do routers adapt to the smaller expert set? Are there any drift in load-balancing metrics?

5- Did you explore post-pruning training to see how the model performs after some lightweight tuning?

---

> ### Author Response · Authors · 2025-11-23
> **Rebuttal for Weakness 1,2,3**
>
> Thank you for your thoughtful suggestions and constructive feedback. Below we provide our detailed rebuttal and clarifications. We will incorporate all corresponding additions—including tables, analyses, discussions, and necessary explanations—into the revised version of the manuscript.
>
> - **W1**: Baseline Coverage
>
> (1) **Hypernetwork-based baselines**. We appreciate this interesting idea. While hypernetworks have not been commonly used for expert pruning, we propose the following implementation plan:
>
> a).Construct training data: For each input in the dataset, we collect the expert activation frequencies. Experts with high activation frequencies are labeled as 1, while others are labeled as 0, forming a binary label vector for each input.
>
> b).Train the hypernetwork: Train a model to predict these expert retention labels from the input data.
>
> c).Apply to pruning: Use the calibration dataset as input to the trained model and retain the experts predicted as 1.
>
> If this approach is suitable, we will conduct these experiments promptly.
>
> (2) **Weight importance baselines**. Following the suggestion, we implemented a weight importance baseline using the Frobenius norm to evaluate each expert's parameters, inspired by [a]. The top-50% of experts by importance score are retained. Results are shown in Table 1.
>
> Table 1. Performance Comparison of Weight Importance (Qwen3-30B-A3B-Thinking, 50% pruned)
> |          | MMLU  | Math500 | AIME25 | LCB   | Average |
> |----------|-------|---------|--------|-------|---------|
> | Original | 78.59 | 96.12   | 85.00  | 66.00 | 81.43   |
> | Weight Importance | 23.76 | 9.20    | 6.70   | 0.00  | 9.92    |
> | Ours     | **52.56** | **95.60**   | **80.00**  | **52.40** | **70.14**   |
> |
>
> We believe these enhancements strengthen our baseline comparisons and thank you for the constructive feedback.
>
> [a] Hao Li, Asim Kadav, Igor Durdanovic, Hanan Samet, Hans Peter Graf. Pruning Filters for Efficient ConvNets. ICLR 2017.
>
> - **W2**: Incomplete related work
>
> We thank the reviewer for this suggestion. Classic works [1,2] indeed adopt greedy, per-layer selection strategies, and we will cite and discuss them in the revision. However, these methods fundamentally differ from our setting: they target dense CNNs with modest layer widths, whereas MoE pruning involves selecting among hundreds of experts per layer in ultra-large language models (e.g., 256 experts in DeepSeek-R1 with more than 600B parameters). Directly applying their layer-wise greedy reconstruction is computationally infeasible in MoE architectures due to the exponential number of expert combinations. Our contribution is precisely to make reconstruction-based, layer-wise pruning scalable for large MoEs by introducing a coarse-to-fine approximation with a theoretical bound, reducing complexity from exponential to polynomial. Thus, while the high-level “layer-wise greedy” idea is shared, our method addresses a fundamentally different scaling challenge and enables a class of pruning algorithms that prior work cannot practically support. We will incorporate these points into the revised version for improved clarity.
>
> - **W3**:Limited task coverage
>
> Thank you for this important suggestion. To better evaluate cross-domain robustness, we have expanded our experiments to include a wider range of benchmarks, as shown in Table 2.
>
> Table 2. Performance comparison across diverse benchmarks on mixed datasets (Qwen3-30B-A3B-Thinking, 50% pruned). GPQA is denoted as GPQA_DIAMOD
> |       | MMLU  | GPQA | BBH   | GSM8K | HumanEval+ | MBPP+ | MBPP  | Average |
> |--------------|-------|------------|-------|-------|----------------|-----------|-------|---------|
> | Original     | 78.59 | 43.43      | 65.67 | 96.06 | 85.37          | 73.28     | 68.80 | 73.03   |
> | weighted     | 47.16 | 21.86      | 63.43 | 42.30 | 0.00           | 0.00      | 0.00  | 24.96   |
> | HC-SMoE      | 43.54 | 22.22      | 62.42 | 69.67 | **51.22**          | 40.21     | 27.40 | 45.24   |
> | Ours         | **52.56** | **32.83**      | **66.66** | **93.93** | **51.22**          | **63.23**     | **50.40** | **58.69**   |
> |
>
> The results show that our method consistently outperforms all baselines across every task, demonstrating stronger generalization and broader robustness.

---

> ### Author Response · Authors · 2025-11-23
> **Rebuttal for weakness 4,5 and questions 1,2,3**
>
> - **W4**: Limited novelty
>
> （1）While reconstruction-loss metrics themselves are not new, prior methods are computationally infeasible for large MoE layers such as DeepSeek-R1 with 256 experts. Our contribution is to make reconstruction-based pruning practically feasible by reducing the complexity from exponential to polynomial through a theoretically justified approximation, enabling an operation that was previously unaffordable on trillion-parameter MoEs.
>
> （2）First, the two methods differ fundamentally in purpose and mechanism. **DoReMi reweights data to guide training**. It modifies gradient updates so the model learns more from beneficial domains. In contrast, **pruning involves no training at all**: our method uses output-discrepancy signals only to decide which expert substructures are better suited to the data. Second, prior pruning works always rely on a fixed calibration set, but **we show that data composition significantly influences which experts should be retained**, and dynamically adapting this mixture substantially improves cross-domain robustness. Thus, the novelty lies not in reusing DoReMi, but in revealing and leveraging the data-dependent nature of MoE pruning to select more appropriate substructures without any weight updates.
>
> - **W5**: Missing inference throughput after pruning.
>
> Thank you for pointing this out. Throughput is an important metric for evaluating inference efficiency. We measured the throughput of Qwen3-30B-A3B-Thinking and the model with 50% expert pruned. The original model achieves 4408.41 tokens/s, while the pruned model reaches 5384.07 tokens/s.
>
> - **Q1**
>
> Thank you for pointing this out. The experts are grouped according to the order in the candidate set. We have noted this in Appendix C of our manuscript.
>
> - **Q2**
>
> Thank you for your suggestion. We have now included the results of other pruning criteria for the mixed datasets in Table 3. As shown in the table, our method outperforms the alternatives across all benchmarks. Specifically, it achieves 52.56% on MMLU (vs. 40.64% for HC-SMoE), 95.60% on Math500 (vs. 37.40%), 80.00% on AIME25 (vs. 16.70%), and 52.40% on LCB (vs. 29.60%). The average performance of our method is 70.14%, demonstrating its overall effectiveness.
>
> Table 3. Comparsion with other pruning methods on the mixed datasets (Qwen3-30B-A3B-Thinking, 50% pruned)
>
> |     | MMLU  | Math500 | AIME25 | LCB   | Average |
> |-----------|-------|---------|--------|-------|---------|
> | original  | 78.59 | 96.12   | 85.00  | 66.00 | 81.43   |
> | Weighted  | 39.05 | 16.20   | 0.00   | 0.00  | 13.81   |
> | HC-SMoE   | 40.64 | 37.40   | 16.70  | 29.60 | 31.09   |
> | Ours      | **52.56** | **95.60**   | **80.00**  | **52.40** | **70.14**   |
> |
>
> - **Q3**
>
> Thank you for the question. Our experiments show that the domain ratio (initially rStar-Coder:OpenR1-Math:C4 = 1:1:1) converged to 2:1:1 quickly and remained stable thereafter. We speculate that the stable convergence may be attributed to the relatively similar distributional characteristics between the math and code domains, which facilitates stable weight adjustment. In future work, we plan to further validate this behavior across more diverse domain configurations.

---

> ### Author Response · Authors · 2025-11-23
> **Rebuttal for question4 and 5**
>
> - **Q4**
>
> Thank you for your questions regarding router behavior after pruning. We have analyzed both expert selection patterns and load balancing using the Qwen3-30B-A3B-Thinking model with 50% of experts pruned (retaining 64 experts).
>
> (1) **For the first question**. To evaluate selection uniformity, we grouped the remaining experts into 8 buckets (16 experts per bucket). The distribution of selected experts, visualized in Figure 4  (see Appendix C) of our manuscript, appears uniform across different layers. We further computed the entropy of the router's output distribution across these buckets. As shown in Table 4, all entropy values exceed 2.9 (maximum 3.0), confirming that expert selection remains highly uniform without over-reliance on a limited subset.
>
> Table 4: Entropy of expert selection after 50% pruning
>
> | Layer Number | 6      | 12     | 18     | 24     | 30     | 36     | 42     | 48     |
> |--------------|--------|--------|--------|--------|--------|--------|--------|--------|
> | Entropy      | 2.9113 | 2.9744 | 2.9295 | 2.9545 | 2.9712 | 2.9482 | 2.9386 | 2.9797 |
> |
>
> (2) **For the second question**. We compared the expert activation distributions (Figures 7-8  (see Appendix C) of our manuscript), which show that the pruned model exhibits a flatter activation pattern. To quantitatively assess load balancing, we calculated Gini coefficients for both models. A lower Gini coefficient indicates better load balance. As summarized in Table 5, the pruned model shows consistently lower Gini coefficients across all layers, confirming improved balancing.
>
> Table 5: Gini coefficients (original vs. pruned).
> | Model    | 6    | 12   | 18   | 24   | 30   | 36   | 42   | 48   |
> |----------|------|------|------|------|------|------|------|------|
> | Original | 0.52 | 0.58 | 0.61 | 0.57 | 0.60 | 0.57 | 0.60 | 0.68 |
> | Pruned   | 0.43 | 0.49 | 0.54 | 0.53 | 0.55 | 0.50 | 0.50 | 0.54 |
> |
>
> - **Q5**
>
> Thank you for this question. As shown in Figure 9  (see Appendix C) of our manuscript,, we conducted preliminary post-pruning fine-tuning using LoRA over 250 training steps. The results indicate a consistent decrease in loss as training progresses. However, this represents an intermediate evaluation, and the current training steps are insufficient to reach full convergence. We are continuing these experiments and will report the final convergence results once the training stabilizes.

---

> ### Author Response · Authors · 2025-11-26
> **Gentle Reminder**
>
> This is a gentle reminder regarding the review for our submission. If there are any points in our paper that remain unclear or if further clarification from our side would help you complete the review, please feel free to let us know. We would be more than happy to assist in making our work easier to evaluate

---

> > ### Comment · Reviewer_wtZA · 2025-11-26
> >
> > I thank the authors for their response and conducting additional experiments.
> >
> > ### W1.1
> > Hypernetwork-based pruning is a commonly used technique for reducing the size of large language models (LLMs) (see [1] for an example). In this approach, a hypernetwork is trained (using differentiable discrete operations) to predict which experts to retain or discard. This is done by freezing the main model and training the smaller hypernetwork on a language modeling objective over a given calibration set. Based on the reviewer's past experience, this method is actually a surprisingly strong baseline.
> >
> > ### W1.2
> > Did the authors follow e.g. [2] to find weight importances? If the baseline is calibration-data independent, that's an unfair comparison. Are the values in table 1 without any fine-tuning the baseline? It doesn't seem to be the case and that would be an unfair comparison as the method proposed in this work is more compute intensive.
> >
> > In general I don't understand why structural pruning baselines were not considered in this work.
> >
> > ### W2
> > These method do fundamentally differ from your setting, but the idea has been used so it should be discussed in the related work with a clear explanation of what makes this work different. I don't see this discussion in the related work now.
> >
> >
> > ### W3
> > Thank you for providing the new results. Where in the revision have these results been added?
> >
> >
> > ### W4
> > Thank you for the clarification. While this is a minor concern, as I understand the distinctions, the purpose of a related work section is beyond simply listing prior work. Its primary role is to provide context, helping readers understand how the current research is positioned within the broader landscape. That contextual framing seems to be missing in the current version.
> >
> > ### W5
> > How are these values calculated? Where are they added in the paper? Since the goal of pruning is inference efficiency, providing inference efficiency results is paramount.
> >
> > ### Q1
> > Does this mean there’s no consideration (or investigation) of whether the grouped experts are actually similar in terms of the information they capture? If the grouping is effectively agnostic to content, then the lack of such analysis is a limitation of the work that should be noted.
> >
> > ### Q2
> > Thank you. Has this table been added to the appendix?
> >
> > ### Q3
> > Thank you. That's a limitation worth noting in the manuscript. That the domains considered here have a great deal of similarity.
> >
> > ### Q4
> > Thank you for this analysis. This is insightful.
> >
> > ### Q5
> > Thank you. I was interested in seeing see whether convergence is faster compared to simple baselines.
> >
> >
> > [1] DISP-LLM: Dimension-Independent Structural Pruning for Large Language Models, Gao et. al, NeurIPS 2024
> >
> > [2] LLM-Pruner: On the Structural Pruning of Large Language Models, Ma et. al, NeurIPS 2023

---

> ### Author Response · Authors · 2025-12-02
> **Response to all weaknesses, Q1 and Q2**
>
> - W1.1&W1.2
>
> Thank you for these insightful comments and for suggesting these important baselines. Our method is designed as a training-free approach that requires only forward passes during inference. For fair comparison, we initially focused on baselines with similar computational characteristics. Structural pruning methods, including hypernetwork-based and element-wise importance approaches, typically require substantial training resources. For instance, element-wise importance baselines requires 24 hours(23h36m33s) on four A800 GPUs(80GB each）, the hypernetwork-based baseline requires approximately 10 hours (9h17m48s) on two A800 GPUs (80GB each), while our complete process finishes in 149.84 seconds using a single A800 GPU.
> In response to your feedback, we have implemented and evaluated both suggested baselines:
> Hypernetwork-based baselines[1]: We adopt the same hypernetwork architecture followed with ReinMax, training paramaters and objective function(p=0.5). The hypernetwork outputs determine which experts are selected in each MoE layer.
> Element-wise importance baselines[2]: We compute element-wise importance scores for each expert and aggregate them via summation to obtain final expert importance.
> As shown in Table 6, our method maintains superior performance while being significantly more efficient:
>
> Table 6. Comparison with Structural Pruning Methods.
>  |          | MMLU | Math500 | AIME25 | LCB  | Average |
> | :------- | :--- | :------ | :----- | :--- | :------ |
> | Original | 78.59| 96.12   | 85.00  | 66.00| 81.43   |
> | Hypernetwork-based | 52.01| 89.20   | 40.00  | 18.30| 49.88   |
> | Element-wise importance | 43.58| 89.20   | 63.30  | 28.70| 56.20   |
> | Ours | **52.56**| **95.60**   | **80.00**  | **52.40**| **70.14**   |
> |
>
> These results demonstrate that our approach achieves better performance while being substantially more computationally efficient than training-dependent structural pruning methods. We have included the experimental results in Appendix D: Comparison with Structural Pruning Baselines.
>
> [1] DISP-LLM: Dimension-Independent Structural Pruning for Large Language Models, Gao et. al, NeurIPS 2024.
>
> [2] LLM-Pruner: On the Structural Pruning of Large Language Models, Ma et. al, NeurIPS 2023.
>
> - W2&W4
>
> We have updated the Related Work section to explicitly discuss these prior methods and clearly explain how our setting and contribution differ from theirs. The revised version now includes this clarification.
> - W3
>
> Thank you for your comment. We have included the additional experimental results in Appendix D: Evaluation on Additional Datasets.
>
> - W5
>
> Thank you for your comment. We evaluate the inference throughput of both the original and pruned models on the C4 dataset using EvalScope on a single A800 GPU. The results have been added to  Appendix C: Load-balancing Analysis.
>
> - Q1
>
> To assess the impact of potential intra-group similarity after grouping, we introduce two baseline strategies: a similarity-based method and a random-based method.  The results show that performance remains stable across these different grouping strategies, indicating that our method is robust and not strongly dependent on the specific similarities within groups. The corresponding experiments and discussion have been included in Appendix C: Grouping Strategies Analysis.
>
>  **Table 7. Performance comparison of different grouping strategies.**
>
> | Method | MMLU | Math500 | AIME25 | LCB | Average |
> | :--- | :---: | :---: | :---: | :---: | :---: |
> | Original | 78.59 | 96.12 | 85.00 | 66.00 | 81.43 |
> | Random | **52.59** | 92.20 | 73.30 | 47.10 | 66.30 |
> | Similarity-based | 52.56 | 95.20 | **80.00** | 51.50 | 69.82 |
> | Order-based | 52.56 | **95.60** | **80.00** | **52.40** | **70.14** |
>
> - Q2
>
> Thank you for your comment. We have included the additional experimental results in Appendix D: Evaluation on Additional Datasets.

---

> ### Author Response · Authors · 2025-12-02
> **Response to Q3,Q4 and Q5**
>
> - Q3
>
> Thank you for raising this important point regarding oscillation behavior. We have conducted further investigation into the conditions under which Algorithm 2 may exhibit oscillations.
> We introduced a perturbation analysis to the update rule, defined as $\frac{\exp(\boldsymbol{\epsilon} \odot \boldsymbol{d}^t)}{\|\exp(\boldsymbol{\epsilon} \odot \boldsymbol{d}^t)\|_1}$, where each element of $\boldsymbol{\epsilon}$ is sampled from a uniform distribution over the interval $(0, 10]$. We then visualized the relationship between the variance of the output differences (under perturbation) and the resulting convergence or oscillation status.
> As shown in Figure 7 (see Appendix C: Convergence Analysis of DCDM), the statistical results indicate that larger variances lead to oscillations, whereas smaller variances promote stable convergence. In our experimental settings, the observed variance remains consistently below the convergence mean, which explains why Algorithm 2 does not exhibit oscillatory behavior in our current implementation.  We have included the additional analysis in Appendix C: Convergence Analysis of DCDM
>
> - Q4
>
> Thank you for your feedback. We’re glad that the analysis was helpful.
>
> - Q5
>
> Thank you for your comment. As shown in Figure 9 in Appendix C, we fine-tuned the pruned Qwen3-30B-A3B-Thinking model using LoRA. The loss curves demonstrate that our method consistently achieves lower loss values compared to other approaches, confirming the preserved potential of models pruned with our method after fine-tuning.

---

### Author Response · Authors · 2025-12-02
**Summary of the Rebuttal History**

Dear Area Chair,

We sincerely thank the AC for the time and effort invested in supporting the ICLR review process. To make the evaluation as clear and efficient as possible, we provide the following concise summary of our revisions and how they address the reviewers’ concerns. Below are the key recurring concerns and our corresponding resolutions：
1. `More baselines`（Reviewer wtZA-W1/Q2, Reviewer RQgE-W1/W3/Q3, Reviewer vzcY-W1/Q1, Reviewer yUmm-Q1）

 We introduced and compared five additional baselines including weight-importance methods[1], hypernetwork baselines[2],  Element-wise importance baselines[3], Data-Centric and Model-Centric Expert-Guided methods [4]. This verified the superior performance of our method across all benchmarks.

2. `More models`（Reviewer RQgE-W4/Q1/Q2/Q3, Reviewer vzcY-W1/W2/Q2）

We added more experiments on DeepSeek-MoE-16B and extended evaluation to models of three scales (16B, 30B, 671B). This verified the robustness and consistent improvements of our method across different model sizes and MoE architectures.

3. `More benchmarks`（Reviewer wtZA-W3, Reviewer RQgE-W1/W3/Q3, Reviewer vzcY-W1/Q1）

We expanded the evaluation to a broader set of nine diverse benchmarks, including: GPQA, BBH, GSM8K, HumanEval+, MBPP+, MBPP, BoolQ, OpenBookQA and RTE. This verified that our method outperforms others on average across this comprehensive suite.

4. `Analysis of method robustness and stability`
- **Grouping Strategies**. (Reviewer wtZA-Q1,Reviewer yUmm-W1/Q1) We assessed multiple expert grouping strategies after pruning. This verified that order-based grouping yields a slight performance advantage.
- **Expert Selection Distribution**.(Reviewer wtZA-Q4) We analyzed the distribution of selected experts across layers. This verified a uniform selection pattern with high entropy, showing no subset preference.
- **Pruning Data Efficiency**.(Reviewer RQgE-W1) We tested pruning effectiveness using different numbers of calibration sequences. This verified that performance plateaus with a very small dataset, confirming data efficiency.
- **Pruning Ratio Robustness**.（Reviewer RQgE-W4/Q1, Reviewer vzcY-W3/Q3）We evaluated our method against baselines across various pruning ratios. This verified its consistent superiority and robustness under different pruning ratios.
- **Reweighting Strategy**. (Reviewer yUmm-Q3)We compared exponential and linear reweighting for dataset sampling. This verified the superior performance of the exponential strategy.
- **Algorithm Convergence**. (Reviewer wtZA-Q3) We introduced a perturbation analysis for the DCDM algorithm. This verified its convergence, associating it with low output variance.
- **Load Balancing**.(Reviewer wtZA-W5/Q4, Reviewer RQgE-W3/Q2) We compared expert activation distributions and Gini coefficients before and after pruning. This verified improved load balancing and a corresponding increase in inference throughput.
- **Post-Training Recovery**. (Reviewer wtZA-Q5) We fine-tuned the pruned model and compared training loss. This verified that our method preserves model capacity more effectively than others.
- **Domain-specific Performance Trade-off**. (Reviewer yUmm-W2/Q2) We added detailed cross-domain vs. single-domain comparison and clarified trade-off principles. And we show that supportive domains benefit from mixing while conflicting domains are better handled with specialized calibration.

5. `Conceptual and Theoretical Clarifications`
- **Related work Clarification**. (Reviewer wtZA-W2) We expanded the Related Work section to include CNN-style layer-wise pruning and recent structured LLM pruning methods. We show that these approaches do not scale to large MoEs with hundreds of experts per layer.
- **Theoretical Clarifications**. (Reviewer RQgE-W2) We clarified that the proof relies only on Lipschitz continuity and the triangle inequality, without any independence assumptions, and we highlighted our novel contribution in establishing a scalable approximation with a provable error bound.
- **Clarification of Novelty**. (Reviewer vzcY-W4, Reviewer wtZA-W4) We added analysis showing that our approximation makes reconstruction-based MoE pruning polynomial-time and scalable; and we show that our discrepancy-driven dynamic calibration, unlike DoReMi’s training-based reweighting, improves cross-domain robustness without any training.

[1] Hao Li, Asim Kadav, Igor Durdanovic, Hanan Samet, Hans Peter Graf. Pruning Filters for Efficient ConvNets. ICLR 2017.

[2] DISP-LLM: Dimension-Independent Structural Pruning for Large Language Models, Gao et. al, NeurIPS 2024.

[3] LLM-Pruner: On the Structural Pruning of Large Language Models, Ma et. al, NeurIPS 2023.

[4] Zhang, Z., Liu, X., Cheng, H., Xu, C., & Gao, J. Diversifying the expert knowledge for task-agnostic pruning in sparse mixture-of-experts. Findings of ACL 2025.

---

### Meta-Review · Area_Chair_SYkE · 2026-01-07

**Summary:**

The submission proposes a technique for MoE pruning that combines calibration with pruning.  Highlighted performance claims are a reduction from exponential to polynomial cost, and 98.9% performance retention on deepseekR1 while reducing the number of experts by 50%.

**Reviewer Concerns:**

Main concerns included missing related work (there are *a lot* of papers on MoE pruning, and many that involve calibration), that experiments were limited primarily to one main setting, and novelty.  I will additionally point out that the exponential to polynomial cost reduction may be overstated, as many existing methods use e.g. greedy approaches that are quite tractable and polynomial time.

**Reviewer Scores:**

A majority of reviewers initially felt that the submission was below the acceptance threshold, with one reviewer giving a score of 6.  The authors were very active in responding to reviewer comments.  In my assessment, novelty remains a concern, and the experiments are less systematic in the time that was available for the rebuttal.  The submission could benefit from a more thorough empirical study in a future submission, as well as a clearer articulation on the claimed novelty and computational and empirical improvements over the extensive literature on the topic.

---

### Decision · Program_Chairs · 2026-01-26

Reject